# Transitional rock glaciers at sea-level in Northern Norway

Karianne S. Lilleøren[1], Bernd Etzelmüller[1], Line Rouyet[2], Trond Eiken[1], Gaute Slinde[1], Christin Hilbich[3]

[1]Department of Geosciences, University of Oslo, Oslo, N-0316, Norway
[2]NORCE Norwegian Research Centre AS, Tromsø, N-9294, Norway
5   [3]Department of Geosciences, University of Fribourg, Fribourg, CH-1700, Switzerland

*Correspondence to*: Karianne S. Lilleøren (k.s.lilleoren@geo.uio.no)

**Abstract.** Rock glaciers are geomorphological expressions of permafrost. Close to sea level in northernmost Norway, in the sub-Arctic Nordkinn peninsula, we have observed several rock glaciers that appear to be active now or were active in the recent past.

10   Active rock glaciers at this elevation have never before been described in Fennoscandia, and they are outside the climatic limits of present-day permafrost according to models. In this study, we have investigated whether or not these rock glaciers are active under the current climate situation. We made detailed geomorphological maps of three rock glacier areas in Nordkinn, and investigated the regional ground dynamics using Synthetic Aperture Radar Interferometry (InSAR). One of the rock glaciers, namely the Ivarsfjorden rock glacier, was investigated in more detail by combining observations of vertical and 15   horizontal changes from optical images acquired by airborne and terrestrial sensors and terrestrial laser scans (TLS). The subsurface of the same rock glacier was investigated using a combination of Electrical Resistivity Tomography (ERT) and Refraction Seismic Tomography (RST). We also measured ground surface temperatures between 2016 and 2020, complemented by investigations using an infrared thermal camera, and a multi-decadal climatic analysis.

We mapped the rock glaciers in the innermost parts of Store and Lille Skogfjorden as relict, while the more active 20   ones are in the mouths of both fjords, fed by active talus in the upper slopes. Several of the rock glaciers cross over both the Younger Dryas shoreline (25 m a.s.l.), and the Early to Mid-Holocene shoreline at 13 m a.s.l. Both InSAR and optical remote sensing observations reveal low yearly movement rates (mm-cm yr$^{-1}$). The ERT and RST suggest that there is no longer permafrost and ground ice in the rock glacier, while temperature observations in the front slope indicate freezing conditions also in summer. Based on the *in situ* temperature measurements and the interpolated regional temperature data, we show that 25   the MAAT of the region has raised by 2 °C since the late 19th century to about 1.5 °C in the last decade. MAATs below 0 °C 100-150 years ago suggest that new rock glacier lobes may have formed at the end of the Little Ice Age (LIA).

These combined results indicate that the Nordkinn rock glaciers are transitioning from active to relict stages. The study shows that transitional rock glaciers are still affected by creep, rock falls, snow avalanches, etc., and are not entirely dynamically dead features. Our contrasting results concerning permafrost presence and rock glacier activity show the 30   importance of a multi-methodical approach when investigating slope processes in the edge zones of permafrost influence.

## 1 Introduction

Permafrost, defined as ground with temperatures remaining at or below 0°C for at least two consecutive years (Van Everdingen, 1998), is widespread in mountain areas. The distribution of mountain permafrost is governed by air temperature, snow cover, shadowing, land cover type, and grain sizes; all of these strongly modulated by topography in high-relief settings (Harris and 35   Corte, 1992; Harris and Vonder Mühll, 2001; Rödder and Kneisel, 2012; Deluigi et al., 2017; Gisnås et al., 2017). The ground thermal regime in general influences gravitational processes, like mass wasting, and is thus an important factor for landscape evolution (Berthling, 2011; Egholm et al., 2013; Hales and Roering, 2009; Hales and Roering, 2007).

Some landforms directly indicate the presence of permafrost, e.g. palsas, peat plateaus and rock glaciers. While palsas form in topographic depressions occupied by mires, rock glaciers are located in sloping terrain with talus, avalanche debris, or 40   morainic deposits. Rock glaciers form when ice-cemented ground starts to creep because of the ice's plasticity and gravity

(King, 1986; Delaloye and Lambiel, 2005; Farbrot et al., 2007; Haeberli, 1985; Berthling, 2011). Since rock glaciers are visual expressions of permafrost, the distribution of both active, inactive and relict landforms is indicative of certain climate types at present, in the recent past, and in the more distant past. Therefore, in 2011, a Norwegian rock glacier inventory of 307 landforms was published (Lilleøren and Etzelmüller, 2011), mainly mapped based on the digital aerial photos available at that time. This mapping was based on visual interpretation of the landforms, i.e. convex shape, clear front slope, signs of surface creep or thermokarst, situated downslope of talus or other debris deposits.and included activity status. The quite few examples of mapped rock glaciers close to sea level are situated in areas that were deglaciated before the Younger Dryas (YD), and where climatic conditions favoured permafrost development outside the Weichselian ice margin during the deglaciation (Andersen, 1981; Sollid et al., 1973). These rock glaciers originate exclusively from talus slopes. They sometimes cross raised shorelines, and they vary in size and shape from small lobate landforms to more developed tongue-shaped or lobate landforms, and were always interpreted as being relict. In Norway, clusters of rock glaciers at sea level are found in the Vesterålen-Andøya area in Nordland, the Kåfjord-Lyngen area in Troms and the north-eastern part of Finnmark towards the Barents Sea (fig. 1; Sollid and Sørbel, 1992; Lilleøren and Etzelmüller, 2011). In the 2011 rock glacier inventory, we interpreted the level of activity based on the available aerial photos at that time, and categorized the landforms as either 'active' (steep front slopes, deep ridges and furrows indicating movement, creep structures), 'inactive' (less clear signs of movement, but with a "fresh" appearance), and 'relict' (extensive vegetation cover, thermokarst structures, no signs of movement) based on the characterization by e.g. Barsch (1996). In the study we present here, we had access to kinematic data of the land surface, and we have therefore used the recent categorization of an International Permafrost Association (IPA) action group on rock glaciers inventories and kinematics. Here, 'active' rock glaciers move more than 0.1 m yr$^{-1}$, 'transitional' between 0.01 and 0.1 m yr$^{-1}$ and 'relict' less than 0.01 m yr$^{-1}$ (RGIK, 2022a). In this study, we sometimes refer to landforms as 'intact' landforms. In these cases, we mean the geomorphological appearance of the landforms (Barsch, 1996). However, with one exception, we do not have any evidence of the ice content of the rock glacier interiors.

While the coastal rock glaciers in other parts of Norway are interpreted as relict based on their visual appearance, partly overgrown with vegetation and with relatively smooth topography, the rock glacier clusters in northernmost Finnmark appear more active with steep fronts, ridges and furrows. If these rock glaciers are active, this would mean that the permafrost distribution in the northernmost coastal areas of Norway are more widespread and situated at lower elevations than what is considered today based on the available models (e.g. Farbrot et al., 2013; Obu et al., 2018; Gisnås et al., 2017). In order to investigate the activity and history of these landforms, we launched a mapping and monitoring program, where we (1) produced detailed geomorphological maps for three focus areas (Ivarsfjorden, Store Skogfjorden and Lille Skogfjorden), (2) analysed mean annual surface velocity of all rock glaciers in the area based on Sentinel-1 Synthetic Aperture Radar Interferometry, and (3) monitored one rock glacier (Ivarsfjord rock glacier, north of Hopsfjorden; 71°N, 28°E; fig. 1), by combining different techniques to document internal structure (geophysical surveys), thermal regime (near-surface temperature data loggers and thermal camera) and displacement rates (repeated Structure from Motion (SfM) images by orthophotos and terrestrial laser scans). Combined, these methods give a comprehensive overview of the ground thermal state and rock glacier dynamics in this coastal, northern setting. We specifically investigate the dynamic state of Ivarsfjord rock glacier and the other rock glaciers in the region. We interpret the formation period and how they have developed over time, and we suggest this area to be an interesting analogue to the development of Svalbard rock glaciers in a warmer future. Thus, this study provides new insights about former and present permafrost distribution in this sub-arctic environment, and how these rock glaciers near sea-level transition in response to a changing climate, both post-deglaciation and more recently.

## 2 Setting

Finnmark is dominated by wide fjords in the coastal areas (west, north and east), and large plateaus in the south (Finnmarksvidda). Between the larger fjords towards north, the landscape of the peninsulas is dominated by steep mountain slopes towards the sea and flat plateaus in the interior areas, normally with elevations below 600 m a.s.l. These plateaus are dominated by exposed bedrock or *in situ* weathered material, and in some areas coarse-grained till.

The bedrock of the Nordkinn peninsula north of Hopsfjorden generally consists of siliclastic, folded metasedimentary rocks from late Mesoproterozoic to early Neoproterozoic (fig. 1c). These rocks are parts of the Kalak Nappe Complex, included in the Caledonian fold belt (Schilling et al., 2014). The bedrock in Ivarsfjorden, Store Skogfjorden and Lille Skogfjorden consists of sandstones and phyllites (NGU, 2022). The area is dominated by phyllitic schists and shales, with quarzitic sandstones as southwest-to-northeast belts in the landscape.

During the Pleistocene, Finnmark was repeatedly glaciated, last by the Fennoscandian Ice Sheet (FIS), where ice streams flowing west and north coalesced with the Barents Sea Ice Sheet (BSIS; Boulton et al., 2001; Dowdeswell and Siegert, 1999; Shackleton et al., 2018; Ottesen et al., 2005). The northern areas, including Nordkinn, were early deglaciated, appr. 14-15 cal kyr BP (Romundset et al., 2011). In lake sediment records, Romundset et al. (2011) find indications of an abrupt regression that occurred between 10.5 and 10 cal kyr BP. In the following period between 10 and 5 cal kyr BP, a transgression occurred, while during the last 5 kyr, the relative sea level has fallen by 10 m. A significant uplift of the coast of Finnmark happened when the Barents Sea ice sheet disintegrated, which occurred prior to the main Holocene uplift (at 17-15 cal kyr BP, Winsborrow et al., 2010). Due to the disappearance of the Barents Sea ice sheet, the shoreline displacement rates (land uplift) in the outer part of Finnmark are about three times lower when compared to similar areas in other parts of the Norwegian coast and the marine limit was reached as early as ca 14.6 cal kyr BP (Romundset et al., 2011).

The climate of this part of northern Finnmark varies between a relatively mild, wet maritime climate at the coast, to a dry continental climate in the interior (Normal period 1991-2020; MAAT from 2.6 °C (coast; Slettnes lighthouse) to -1.3 °C (interior; Karasjok); NCCS, 2021). The mean annual precipitation (1991-2020; MAP) ranges from slightly above 600 mm at the coast to 400 mm at Finnmarksvidda, and the snow depth decreases from 200 cm at the coast to 50 cm in the continental areas (Saloranta, 2012b; NCCS, 2021).

The present lower limit of permafrost in continental parts of Finnmark is situated at 500-600 m a.s.l (Farbrot et al., 2013, Gisnås et al 2017), while it increases to 1000 m a.s.l. towards the coastal areas (Farbrot et al., 2008). The snow distribution and elevation have a major influence of the permafrost distribution, as discussed in several studies modelling mountain permafrost in Norway (Farbrot et al., 2013; Obu et al., 2018; Gisnås et al., 2017). The Nordkinn peninsula is at the edge of the modelled regional permafrost extent.

The rock glaciers in Finnmark have previously all been interpreted as relict landforms, indicative of former existence of permafrost (Lilleøren and Etzelmüller, 2011; Lilleøren et al., 2013). Similarly, the large number of frost polygons in Eastern Finnmark are interpreted as relict (Svensson, 1962, 1992; Malmström and Palmér, 1984). These frost-fissures are located both at raised beaches, up to about 100 m a.s.l., and in blockfields at higher elevations (Svensson, 1986, 1962; Malmström and Palmér, 1984; Fjellanger et al., 2006). When excavating relict polygons on raised beaches, ice-wedge casts have been identified, which is a clear indication of former permafrost (Svensson, 1986).

In this study, we have focused on three areas: Ivarsfjorden (north of Hopsfjorden), Store Skogfjorden and Lille Skogfjorden (both south of Hopsfjorden, fig. 1). All three areas are tributary fjords to the main Hopsfjorden system.

## 3 Methods and data processing

### 3.1 Regional geomorphological and kinematical mapping

The main objective of the geomorphological mapping was to identify rock glaciers and other related landforms with similar surface structures, and how they relate to especially relict raised shorelines in the area. For this work we used a high-resolution LiDAR-based Digital Elevation Model (0.25-0.5 m resolution), in combination with aerial orthophotos (1529C10_1529C12 (1975), 7594B7_7594B9 (1982), 11418C10_11418C12 (1992); Norwegian Mapping Authorities) within a Geographic Information System (GIS) environment (ArcMap (© ESRI)).

For the documentation of rock glacier kinematics, we analysed ground surface velocity maps processed with Synthetic Aperture Radar Interferometry (InSAR) based on 2015–2020 Sentinel-1 Interferometric Wide Swath images (tracks 51 and 124). We combined two datasets processed with different InSAR techniques to take advantage of complementary detection capabilities. InSAR measurements are available through the InSAR Norway ground motion mapping service (www.insar.ngu.no; Dehls et al., 2019). The dataset is based on a Persistent Scatterer Interferometry (PSI) technique (Ferretti et al., 2001). InSAR Norway covers the whole country with a ground resolution of approximately 5 x 20 meters (5 meters in the east-west and 20 meters in the north-south direction). The dataset is typically designed for mm $yr^{-1}$ to cm $yr^{-1}$ ground surface velocities, originally developed for investigating movement on infrastructure and large rock slope instabilities (Vick et al., 2020). PSI may fail over fast and non-linear moving areas. To complement InSAR in these areas, additional velocity maps have been processed by averaging exclusively image pairs with a short temporal interval (6 to 48 days). The technique, so called InSAR stacking (Sandwell and Price, 1998), is less robust for low velocities due to remaining atmospheric effects, but allows for documenting high velocities following the methodology detailed in Rouyet et al. (2021). The spatial resolution of the final product is lower than PSI (40 x 40 m). For rock glaciers located on west-facing slopes, we used exclusively InSAR data based on descending radar geometry. To take advantage of the complementary detection capabilities of the two InSAR methods, the final InSAR map is a composite product consisting in the overlap of the PSI results (for values ranging from 0.1 to 30 cm $yr^{-1}$) and the stacking classified velocities (for values ranging from 3 to 100 cm $yr^{-1}$). It does not provide an exhaustive and absolute documentation of the velocity but gives a general overview of the ground dynamics in the study area. The semi-quantitative information is sufficient for categorizing the rock glacier kinematics (order of magnitude of the creep rate), as recommended by the IPA Action Group on rock glacier inventories and kinematics (RGIK, 2022b). The InSAR values correspond to mean annual ground surface velocities corresponding to sensor-to-ground distance change along the radar line-of-sight (LOS). As the view angle is mostly aligned with the slope orientation and that the kinematic analysis remains semi-quantitative, no projection has been applied for this study.

### 3.2 Ivarsfjorden rock glacier

Three series of aerial photos were used to generate DEMs and orthophotos (1975, 1982 and 1992; Norwegian Mapping Authorities). We used the photogrammetric software suite "ImageStation" for the processing of historical data (Hexagon Geospatial Company). From 2016 to 2019, we collected annual aerial photos from the rock glacier and its close surroundings, using a Camflight C8 Rotor Wing Uncrewed Aerial Vehicle (UAV). The camera used was Nikon Coolpix A, with a resolution of 4928 x 3264 pixels, a focal length of 18.5 mm (Sundheim & Andresen 2016), and a resulting ground sample distance (GSD) of ca. 3.5 cm. The images retrieved by drones were processed using the AGISOFT Photoscan software, georeferenced and ortophotos made with 5 cm resolution. 2019 photos were not suitable for further analyses, and discarded (the GNSS reference point had been destroyed and ground control similar to previous years could not be established). We also acquired terrestrial laser scans (TLS) in the years 2017-20, using a Riegl VZ1000 terrestrial scanner, covering most of the surface. The DEMs generated from the TLS have a ground resolution of 2-15 cm and an accuracy of 2-4 cm (Table 1).

The multi-temporal DEMs were subsequently analysed for vertical changes over the rock glacier body, by subtracting the newest from the oldest DEMs between selected periods. For horizontal displacement analysis of the rock glacier, we used the Correlation Image Analysis Software (CIAS; Kääb and Vollmer, 2000; Heid and Kääb, 2012). The software uses the orthophotos to recognize objects such as large stones and blocks on all images, and then calculates the coordinate displacements of the objects. We analysed surface displacements between 7 image pairs between 1975 and 2018.

Electrical Resistivity Tomography (ERT) and Refraction Seismic Tomography (RST) were carried out during the field seasons of 2017, 2018 and 2019 (fig. 1). ERT documents the electrical resistivity distribution of the subsurface by injecting a current between two electrodes, and measuring the resulting electrical potential differences between two other electrodes along the profile. The depth of investigation depends on the distances between the current electrodes employed along the profile and the profile length, with larger distance giving greater penetration. Liquid water in the ground (soil moisture, ground water) causes low electrical resistivity values, whereas the resistivity of the same material can increase strongly under frozen conditions (Hauck and Kneisel, 2008). As ice acts as an electrical insulator, the resistivity increases with increasing ice content, and high electrical resistivities can indicate frozen conditions (but also dry porous material, as air is also an electrical insulator). Seismic tomography documents the P-wave velocity distribution along the profile by emitting seismic waves at several shot points and measuring the resulting travel times between source (hammer) and receiver (geophones). The P-wave velocity is a function of the elastic properties of the ground material, and the analysis of the seismic travel times provides structural information about the different subsurface layers. Because of their complementary nature, ERT and seismic refraction are often combined for permafrost applications to distinguish between ground ice (high resistivity and medium P-wave velocities), liquid water (low resistivity and P-wave velocities) and air (high resistivity, low P-wave velocities; Hilbich, 2010). Further, the obtained specific resistivity and P-wave velocity distributions can be used as input variables in a petrophysical model approach to quantify the volumetric fractions of the four phases (ice, water, air and rock) in the ground under the assumption of a site-specific porosity distribution (Hauck et al. 2011). This model approach has previously successfully been applied to various permafrost occurrences (e.g. Pellet et al. 2016, Mewes et al. 2017, Halla et al. 2020).

For ERT, we used an ABEM Terrameter LT, with 2 or 4 cables, and an electrode spacing of 2 m. For results' inversion, we used the Res2DINV software (Aarhus Geosoftware; Loke, 2018; Loke and Barker, 1996a, b). For RST, we had a Geode hammer seismograph (© Geometrics) with 24 geophones. We used a geophone spacing of 4 m. In 2017, we measured two short separate ERT profiles (80 m, 2 cables), one close to the centre of the rock glacier and the other at the front crossing into the rock glacier forefield ('ERT 2017', fig. 1d). In 2018, we measured the full rock glacier length by ERT, a total of 480 m ('ERT 2018', fig. 1d), while in 2019 we did seismic surveys along two major profiles ('RST' in fig. 1d). A weakness in our methodology is that we were not able to apply the ERT and the RST measurements at the same time, although we are aware that ground conditions may change over time.

In 2015, we employed 15 miniature data loggers within and outside the Ivarsfjord rock glacier boundary (fig. 1d). We used MAXIM iButtons, with an accuracy of ±0.5 °C, and placed them in open voids protected by small cairns at the surface. We received full-year temperature data from most of the loggers between 2015 and 2020. To identify air circulation in the rock glacier, we investigated the rock glacier front using a thermal camera (Teledyne FLIR C3) measuring infrared radiation. On one day in September 2018, we took ca. 20 pictures distributed along the front, with a 1 m distance between the camera and the object. The thermal camera has an accuracy of ±2 °C.

To place the investigated rock glacier in a climatic context and to study the historical development of the temperature in the study area, we extracted temperature data from a gridded climate data set (daily air temperatures and precipitation) available for all of Norway since 1957 at a ground resolution of 1 km. This dataset, in the following called "*SeNorge*", was established by interpolation between meteorological stations (Lussana et al., 2018; Saloranta, 2012a), and is updated daily. To evaluate temperature development since the end of the Little Ice Age (LIA) at the study sites, we followed two strategies. First, we adapted the *SeNorge* data series back to 1957 at Ivarsfjorden, and made regression analyses between each of the

Ivarsfjorden ground surface temperature and the *SeNorge* daily air temperature series ($R^2 > 0.8$). The difference between ground surface and air temperature occurs especially in winter because of surface snow cover. Second, we applied a linear regression between the *SeNorge* data and the observed temperatures at Vardø radio meteorological station ($R^2 = 0.96$), where continuous observations of air temperatures exist since 1868.

## 4 Results

### 4.1 Hopsfjorden Quaternary geology and geomorphology

Mapping and field observations in the Hopsfjorden area distinguished between rock glaciers appearing with and without visible signs of movement ('intact' and 'relict' rock glaciers, respectively, following Barsch (1996)). Relict rock glaciers are situated in the southern and innermost parts of the fjords, and the intact landforms are situated towards north and close to the mouths of the fjords (figs. 2b and 2c). There are several differences between these inner and outer systems, both in terms of geomorphology and geology. The intact rock glaciers are located below steep and high headwalls, where rockfall debris dominate, and blocky talus cones are common. The lithology of these rock glaciers are quarzitic sandstones. The rock glaciers interpreted as relict are located in gentler slopes with lower headwalls, and affected by both rockslide debris, soil creep and mudslides. These rock glaciers consist of a mix of phyllite and more coarse-grained debris. The transition from intact to relict rock glaciers has a quite distinct spatial distribution, where the southern parts of Store and Lille Skogfjorden exclusively contain relict rock glaciers, while the northern parts have both relict and intact rock glaciers. However, all observed rock glaciers in the area, whether they are intact or relict, are found in the northwest-facing eroded slopes of the sandstone belts trapped between the phyllites. The slopes have a dominantly western exposure and elevations between 20 and 100 m a.s.l. (fig. 2). The slope systems are generally complex including a wide range of processes, from solifluction to rock falls feeding into talus slopes, and landslides reaching the shoreline. Several of the processes are superimposed, like solifluction on an old landslide, or frost creeping talus cones.

The marine sediments in the mapped areas reach an elevation of 60 m a.s.l. in Hopsfjorden. The thickness is unknown, and the surface is characterized by multiple old shorelines (fig. 2). The uppermost marine sediments were formed during the first deglaciation and the following regression of the sea level that occurred after 15k cal yr (Romundset et al., 2011). The later Holocene transgression is visible as a large, distinct but discontinuous ridge, at about 13 m a.s.l.

An interesting observation in Store Skogfjorden was a rockslide that occurred at some point between 2008 and 2018, when we have aerial photos (fig. 3). This rockslide was released in the sandstone zone, had a short transport length (up to 200 m), covered an area of ca. 0.08 $km^2$ and deposited a volume of ca. 1200 $m^3$ (assuming an average thickness h ≈ 15 m). The deposit is shaped as a lobe with a steep and distinct front slope, showing surface structures that could be interpreted as creep features, and thus be misinterpreted as a rock glacier.

Large fields of semi-regular networks of linear depressions are observed on the blockfield-covered mountains between Lille and Store Skogfjorden and in the Sandfjellet mountain, east of the Ivarsfjord rock glacier (fig. 2a). Similar patterns have been observed and described other areas of Finnmark (e.g. Fjellanger et al., 2006). Excavations in an interception of two such furrows at Buhkkačearru (450 m asl), Varanger peninsula (east of our study area), revealed a 1-1.5 m deep feature interpreted as a relict ice-wedge (Malmström and Palmér, 1984). Malmström and Palmér (1984) found no evidence of recent permafrost during their fieldwork.

### 4.2 Regional InSAR-based kinematic analysis

From the analysis of InSAR mean annual velocity at the regional scale, clear kinematic patterns are identified. In general, especially the upper talus parts or upper small, superimposed rock glacier lobes are moving with significant ground

displacement rates between 3 cm yr$^{-1}$ and ~30 cm yr$^{-1}$ (fig. 4A). The flatter lobate parts between the feeding zones and the fronts display small to negligible velocities generally below 1-3 cm yr$^{-1}$ (fig. 4). The rockslide deposit referred to in the previous section (fig. 3A) also shows considerable movement rate above 10 cm yr$^{-1}$. In the regional survey, the Ivarsfjorden rock glacier shows the same displacement pattern as the rock glaciers in Store and Lille Skogsfjord. InSAR velocity maps are used to categorize the order of magnitude of the rock glacier creep rates (fig. 4B), according to the new recommendations of the IPA Action Group on rock glacier inventories and kinematics (RGIK, 2022b). Based on the geomorphology and the InSAR kinematics, 44 rock glacier units are identified and indicatively delineated (fig. 4B). A kinematical attribute is associated to each unit. Most units (27) have very slow creep rates (mm yr$^{-1}$). 13 units have intermediate kinematic attribute (mm-cm yr$^{-1}$), while only a few (4) have cm yr$^{-1}$ or cm-dm yr$^{-1}$ annual rate.

### 4.3 Ivarsfjorden rock glacier monitoring and analysis

We estimated the lowest accuracy for any of the DEMs derived from the orthophotos to ±0.41 m, thus all difference values below 0.5 m were discarded (Table 1). The elevation differences between the various DEMs on the rock glacier are primarily in the range of ±2 m, which correspond to vertical changes of 0–30 cm yr$^{-1}$ (Table 2, fig. 5). The most persistent changes over time are observed at the talus cones or small lobes in the upper parts, in the front slopes and in some lobes in the centre of the rock glacier (fig. 5).

When comparing the elevation differences between the various periods, some patterns can be highlighted. In all periods, the talus cones increased in elevation, i.e. accumulated mass, in the upper part of the rock glacier at rates of 5-20 cm yr$^{-1}$. The 1982-2017 and 1992-2017 periods also show a mass accumulation at the front of the rock glacier at 6-10 cm yr$^{-1}$ (Fig. A1, in appendix), where the largest rate of increase occurs in the 1992-2017 period (Table 2).

The elevation differences between the 2016 and 2017 UAV DEMs were larger than the previous years, in general in the range of ±5 cm yr$^{-1}$, mostly due to surface raising (fig. 5).

We also compared TLSs from 2017 and 2020 which revealed lower yearly rates than between 2016 and 2017, but generally a lowering of the surface of mostly between 0 and 10 cm yr$^{-1}$ (fig. 5).

The analysis of block movement on the rock glacier revealed significant horizontal velocities of generally around 0.5 to 1 cm yr$^{-1}$ between 1975 and 2017 (fig. 5). Maximum velocities were measured in the northern area of the rock glacier, close to the talus/avalanche cone, with values between 2 and 3 cm yr$^{-1}$. Comparison of drone-based orthophotos between 2016 and 2017 shows a more evenly distributed pattern, with higher displacement rates than the previous time periods. The velocities correspond well with the regional InSAR observation shown in fig. 4.

The long ERT profile (2018) provides measurements from the rock glacier forefield and into the upper talus cones feeding the rock glacier (fig. 6). The resistivity profiles show a distinct transition from the area outside the rock glacier (< 5 kΩm) and the rock glacier body (>> 10 kΩm), with maximum values of > 100 kΩm. High resistivity values in such blocky material are normally related to the fact that pores between the blocks are either filled with air or ice. This system can be interpreted as ice and frozen ground below a thawed active layer.

The highest velocities recorded in the seismic refraction profiles were just above 1500 m s$^{-1}$ at the lowest depth of penetration. The majority of recorded velocities was in the range of 500 to 1000 m s$^{-1}$, which are the typical velocities expected from unconsolidated debris with large air pockets (fig. 6). In the case of substantial ice content in the pores, we would expect velocities well above 2000 m s$^{-1}$ and for pure ice above 3000 m s$^{-1}$ (Hauck et al., 2011). Frozen rocks typically give a seismic velocity of 3500-4000 m s$^{-1}$. The wave velocity increases with depth, and this is probably due to debris compaction.

In summary, the results of the two geophysical methods seem to contradict each other: while the ERT data indicate frozen conditions (high resistivities), the RST data do not detect any probability for ground ice. This paradox has already been observed in other situations (e.g. talus slopes), where ground ice can be expected, but with limited volumetric ice content,

which does not sufficiently affect the seismic p-wave velocity. We therefore interpret the data in that way, that the presence of ground ice cannot be excluded from the interpretation of both geophysical methods, but that the RST data strongly suggest a small overall ice content (little saturation of the available pore space).

The measured ground surface temperatures show mean annual values between 1.8 and 3.6 °C from 2015 to 2020 (fig. 1d). The lowest temperatures were recorded (1) in the upper slope of the rock glacier (2.5 °C in average for 3 loggers), (2) in the northern front edge of the rock glacier where we also observed gusts of cold air in summer (2.5 °C), and (3) in one of the creeks escaping the rock glacier (2.6 °C). The modelled *SeNorge* air temperature of Ivarsfjorden rock glacier (grid cell mean elevation 116 m a.s.l.) of the same time period is 1.7 °C, slightly lower than the measured mean annual ground surface temperatures.

We complemented our thermal analyses with infrared pictures taken in the front slope of the rock glacier (fig. 8). On an unusual warm day in September 2018 (> 20 °C), it was possible to feel gusts of cold air escaping the lowest parts of the rock glacier, and the thermal camera showed areas of 0 °C in parts of the front slope network of blocks and air that day (fig. 8).

## 5 Discussion

### 5.1 Rock glaciers in Hopsfjorden – active today?

One of the major aims of this study was to evaluate the activity of the rock glaciers in the area. If active, this would imply permafrost conditions at sea level in northern Norway, following the rock glacier definitions in Haeberli (1985), Barsch (1992) and Berthling (2011). Both current climate information and permafrost models suggest that these coast-near areas are permafrost-free (Gruber, 2012; Gisnås et al., 2013; Westermann et al., 2013; Obu et al., 2018). However, landforms such as palsas and peat plateaus are found in mires developed close to sea-level, especially in glacio-fluvial delta deposits, all along the northern coasts of Finnmark (Sollid and Sørbel, 1998; Borge et al., 2017; Meier, 1987; Kjellman et al., 2018), which clearly demonstrate sporadic permafrost at these locations. Both the peat cover associated with organic material and the blocky talus material normally depress and delay warming of ground temperatures, and thus both palsas and rock glaciers can be found below the regional lower limit of mountain permafrost, such as in high latitude mountain areas in e.g. Scandinavia and Iceland (King, 1986, Delaloye and Lambiel, 2005, Farbrot et al., 2007a). An active rock glacier in a permafrost environment should move, and the movement should be related to the deformation of internal ice bodies (Berthling, 2011).

Based on the yearly displacement rates from optical remote sensing in the Ivarsfjord case study and from InSAR at the regional scale, we see that there is a systematic pattern in the displacement of the slopes inhabited by rock glaciers. Most of them have the maximum displacement values in the upper slopes and the frontal slopes, while only some rock glaciers have displacement rates of more than 3 cm yr$^{-1}$ over their whole area. Although the absolute values of the displacement may differ slightly between the methods due to different time periods, resolutions, and measurement dimensionality, the overall pattern is similar and comparable. The movement values are in the mm to cm yr$^{-1}$-range. In comparable topography, rock glaciers terminating on strandflats in western Svalbard had velocities around 1-5 cm yr$^{-1}$ according to GPS measurements (Berthling et al., 1998; Farbrot et al., 2005), and InSAR (Bertone et al., 2020). Svalbard lies in the continuous permafrost zone, and the low movement rates for the landforms ending on strandflats are partly attributed to low-inclined slopes (Berthling et al., 1998). Therefore, low displacement rates do not necessarily exclude active permafrost creep.

However, rock glaciers investigated in other marginal permafrost areas in Europe, such as the Pyrenees or the Carpathians, show both similar and different flow fields as the Ivarsfjord rock glacier. In the Pyrenees, Serrano et al. (2006) observed high surface displacement rates (30-60 cm yr$^{-1}$ along the central flow line) on an active rock glacier in an area with MAAT close to 0 °C. This is in accordance with observations from e.g. the Alps where degrading permafrost and increased

temperatures cause the rock glaciers to accelerate (e.g. Kääb et al., 2007). As a contrast, active rock glaciers in the southern Carpathians also in an area of MAAT close to 0 °C, have been observed to move very slowly (typically up to 1.5 cm yr$^{-1}$), interpreted as a consequence of a thin ice-rich deforming layer (Necsoiu et al., 2016). This latter study also reports very similar flow patterns in the areas surrounding the rock glacier compared to the rock glacier itself, suggesting a generally active environment with other periglacial slope processes such as solifluction.

The ground resistivity measurements on Ivarsfjorden rock glacier showed resistivity values above 50 kΩm, increasing up-slope, which would indicate possible high ice content. However, the RST surveys revealed velocities in the zone of the resistivity maximum far below what to be expected of massive ice, with values close to or below 1000 m s$^{-1}$, which rather would imply a porous air-filled medium such as blocky talus deposits (Hauck et al., 2011). Deeper in the ground, the velocities reach up to 2000 m s$^{-1}$, which would be more probable for permafrost, but can also be interpreted as compacted talus deposits (the depth perfectly agrees with the depth of the third layer in the ERT). This clearly weakens the permafrost presence hypothesis. An uncertainty is that the two methods were utilized in two consecutive years (ERT in 2018 and RST in 2019), which could imply that the ground thermal regime was different during the two investigations. Temporary ice layers could have formed and resulted in the high resistivity layers measured in 2018. However, 2018 was a summer warmer than average and 2019 was colder than average. We therefore consider that this alternative is unlikely.

The GST monitoring clearly showed annual average temperatures above 0 °C at all places, with a temperature range of ca. 2 °C. Such variations over short distances are commonly observed in mountainous areas (Gubler et al., 2011), and is attributed to snow variations, topographic shadowing, and variations in material properties including grain sizes (Gisnås et al., 2016). The GST observations contradict the existence of extensive permafrost in the rock glacier, although we may see some indications of a colder rock glacier surface towards the upper talus slope in Ivarsfjorden rock glacier (figs. 1D and 8). The thermal camera recorded slightly negative summer temperatures in the front slope of the Ivarsfjord rock glacier, perhaps a sign of a chimney effect causing the dense, cold air to sink through the openwork blocks of the rock glacier (e.g. Lambiel and Pieracci, 2008; Wicky and Hauck, 2017; Kenner et al., 2017; Yuki et al., 2003). We never observed ground ice during fieldwork. However, the present cold air flow indicates the existence of at least minor ice bodies in the rock glacier.

In summary, the thermal camera imagery and the ERT measurements suggest favourable conditions for permafrost occurrence, but the results based on RST and *in situ* temperature loggers tend to contradict this conclusion. The chimney ventilation effect probably cools the ground in summer, and also indicates that there is an open subsurface network to support air flow. Our measurements were generally performed at the end of the melting season, in late summer and early fall, thus we did not observe warm air escaping from the upper slopes of the rock glacier during winter. There might also be an ice core present at one or several locations in the ground, that grows in winter and that mostly disappears during summer (Delaloye and Lambiel, 2005). If so, we would suggest that this ice body is situated at the upper rock glacier, bordering the talus slope.

The deformation rates that are observed both from the InSAR and optical remote sensing analyses is therefore most probably not caused by permafrost creep, driven by massive deforming ice bodies, at least in most of the landforms with a similar movement pattern to Ivarsfjorden rock glacier. This is supported by the fact that most movement is observed in the talus feeding the rock glaciers, and may be attributed to processes other than ice deformation such as solifluction. This observation also suggests that kinematic data to document surface movement should be used with caution for defining activity and inferring information about the ground ice content of the rock glaciers. We support the definition of the activity as exclusively referring to the efficiency of sediment conveyance (expressed by the surface movement) without any inference on the ice content (RGIK, 2022a). Our investigations generally support the conclusion that a documented creep rate is only one piece of information among others to categorize a rock glacier and should be complemented by other geomorphological criteria. Knowledge of rock glacier ice content is also crucial for a comprehensive characterization of the rock glacier state, but has to be treated independently to its kinematics.

## 5.2 Development of the Hopsfjorden rock glaciers

The observations from Hopsfjorden discussed above can be interpreted as landforms in transition from an active to a relict state in response to climate change and atmospheric warming. The landforms must have been formed during periods with cooler climate and a favourable topographic and geological setting.

The location and existence of the rock glaciers in Hopsfjorden are clearly controlled by the local geology, as they are almost exclusively found in the belts of quarzitic sandstones. The rock glaciers in this area all have a westerly aspect, which

can be related to the foliation of the bedrock in the area. Considering the slope aspect, and hence solar insolation, there should not be any difference between the phyllite and the sandstone slopes; the difference is interpreted to be in the weathering products of the two dominating bedrock types. It is well known that phyllite-type bedrock is not a common source for rock glaciers (Haeberli et al., 2006; Ikeda and Matsuoka, 2006). Phyllites or similar schist types produce more fine-grained weathering material, which is frost-susceptible, and more prone to slow slope movement such as solifluction or episodic rapid

events such like debris flows (Haeberli et al., 2006; Matsuoka and Ikeda, 2001). The higher competence of the quarzitic sandstone, on the other hand, produces boulders that alter the air circulation and the ground temperature, similar to areas of blockfields. The physical appearance of the rock glaciers are affected by how coarse the rock glacier material is, where blocky rock glaciers have sharp frontal edges and multiple ridges, while the pebbly or more fine-grained rock glaciers have subdued frontal slopes and often lack the ridge and furrow systems (Matsuoka and Ikeda, 2001; Ikeda and Matsuoka, 2006).

Rock glaciers develop over long time periods (millennia), fed by low magnitude and high frequency events below rock walls. However, certain types of rapid events such as snow avalanches (Humlum et al., 2007) or rockslides (Etzelmüller et al., 2020) may form similar-shaped landforms. The observed rockslide south of Store Skogfjorden (fig. 3) is interesting for understanding which processes dominate the build-up of scree in Norwegian slopes. The sandstones of the release zone favour production of large boulders, which is favourable to chimney effects and cooling of the ground in summer. This rockslide

demonstrates that the slopes in the area are unstable due to fault zones, where the main lines of strike are subparallel to the length direction of all the tributary fjords of Hopsfjorden. Rock glaciers are defined as an accumulated mass of ice-cemented, but unconsolidated debris (Barsch, 1992; Berthling, 2011), but permafrost creep can also occur as secondary processes in all kinds of loose material deposited in slopes in cold climate areas. This landslide deposit could, if situated in a permafrost environment, develop into a rock glacier over time, such as is suggested, for example, for some large debris bodies in Iceland

(Etzelmüller et al., 2020) or as a paraglacial response to deglaciation (Mccoll, 2012; Ballantyne, 2002). Therefore, the other rock glaciers in the same fjord system, situated along the same fault line, could also have developed from the deposits of one or more low frequent and high magnitude rockslide events. From the InSAR deformation rates we see that this rockslide deposit still moves at a velocity above 10 cm yr$^{-1}$, indicating that the stabilization time for such unconsolidated material is rather long, or that creep processes quickly start to rework the deposited mass.

Some of the rock glaciers cross several raised shorelines, for example one particular rock glacier at the mouth of Lille Skogfjorden (fig. 2c). This rock glacier must therefore have been active during and following the land uplift of the early Holocene. The most prominent shoreline located by this rock glacier might be the Early to Mid-Holocene shoreline (Tapes) connected to the transgression sea-level rose faster than the vertical uplift of the crust, which according to Sollid *et al.* (1973) is situated ca. 13 m a.s.l. This is consistent with our observations. The paraglacial relief of the slopes may itself have caused

rock debris masses to form and further to creep in a permafrost environment (Mccoll, 2012; Ballantyne, 2002). The early deglaciation (14-15 cal kyr BP; Romundset et al., 2011) and no later glacier advances in the area left enough time for proper rock glacier accumulation over millennia, with probable stages of less or no activity during the Holocene, e.g. during the Early Holocene Thermal Maximum (HTM), and the later Roman, Medieval and Modern Warmings.

The rock glaciers may have had several active phases, and colder time periods such as the Neoglaciation and the LIA

could have formed several generations of observed rock glacier lobes. The phenomenon of newer lobes forming on top of older have been observed other places, like in the Alps (e.g. Kellerer-Pirklbauer et al., 2012; Armschwand et al., 2021) or in

Iceland (e.g. Kellerer-Pirklbauer et al., 2008). Our study area has warmed by about 2 °C between 1868 and present, and the same between 1957 and present (fig. 7). With an assumed MAAT of 1.6 °C in the 2010-2019 decade, the MAAT could have been just below 0 °C both in the middle of the last century and the end of the 19[th] century. This temperature raise could have triggered a change from an environment where permafrost was sporadically present to an environment with thawing and non-favourable conditions for permafrost. Landslides released prior to the 20[th] century could therefore develop some permafrost, and further some creep, while the presumed permafrost presence is degrading under the current climate. It is however doubtful that completely relict landforms prior to the LIA could have reactivated because of this temperature decrease.

**5.3 Finnmark rock glaciers: an analogy to Svalbard in a changing climate?**

Nordkinn rock glaciers differ from most other rock glaciers in mainland Norway by terminating close to sea level on a flat coastal plane, comparable to near-shore rock glaciers described from western and northern Svalbard (Berthling et al., 1998; Farbrot et al., 2005; Sollid and Sørbel, 1992; Liestøl, 1961), where rock glaciers creep from talus slopes onto the strandflat. Based on earlier studies, these Svalbard landforms were marked as moraine deposits on geomorphological maps, but Liestøl (1961) introduced the term "talus terraces" for footslope landforms *"built of ice and rock debris from the slope above (...)"* (Liestøl 1961, p. 102). Liestøl (1961) acknowledges that the "talus terraces" at numerous locations resemble rock glaciers, and they would indeed fall into the present-day rock glacier definition. The rock glaciers in western Svalbard are creeping into the strandflat with relatively low velocities, mostly below 10 cm yr[-1] (Berthling et al., 1998; Isaksen et al., 2000), attributed to cold ice and low inclination as they run out over the strandflat (fig. 9). Despite difference of climate settings, both for the Hopsfjorden and western Svalbard rock glaciers, there is a zone of higher displacement rates observed in the talus directly above the rock glacier, i.e. at places where the slope angle starts decreasing.

For Svalbard, the air temperatures fluctuated strongly over the 20[th] century (fig. 8), with an overall increasing trend of +0.3 °C/decade for the Svalbard airport meteorological station (Nordli et al., 2014). Between 1971 and 2017, the temperature increased at the same meteorological station by +1 °C/decade (Hanssen-Bauer et al., 2019). During the normal period 1961-1990, the average annual air temperature at Svalbard Airport was -6.7 ˚C. It had increased to -4.6 ˚C for the 1981-2010 period. The strongest increase in temperature occurred during winter (+3.5 ˚C) in contrast to the summer temperature (+1 ˚C). This pronounced increase in temperature during the winter is a pattern observed at all weather stations in Svalbard. The increase in air temperature is expected to continue throughout the 21st century, where different models imply an increase in the range of +2.8 ˚C to +7.8 ˚C (Førland et al., 2011). The air temperature increase was considerable lower in Northern Norway, with values of +0.1 °C/decade since 1901, but +0.5 °C/decade between 1971-2000 and 1985-2014 (Hanssen-Bauer et al., 2015). According to the CryoGRID 1.0 model, approximately 27 % of the land surface area of Northern Norway was underlain by permafrost in the period 1961-1990 (Farbrot et al., 2013). This area was reduced to 19 % for the period 1981-2010, due to a temperature increase over the last decades. The degradation of permafrost and increase of temperature are expected to continue in the next decades (Farbrot et al., 2013).

The landform resemblance between Svalbard and Hopsfjorden both in assemblage and velocities is quite striking. However, in the current framework, conditions are not comparable. In Svalbard there is deep, cold and widespread permafrost, while in Hopsfjorden there is at most isolated patches of permafrost still present. The velocities of the Svalbard rock glaciers are probably low because of low ground temperatures, and hence low ice plasticity, and gentler slopes on the strandflat. On the other hand, the low velocities of the Hopsfjorden rock glaciers are probably because of little pore ice left in the system. Rock glacier velocities tend to speed up when warmed (Kääb et al., 2007), and in this area, it seems that this stage in the development is passed. However, temperatures in Hopsfjorden during the LIA are probably comparable to the Western Spitsbergen temperatures today. In this way, the Hopsfjorden area can serve as a climatic and geomorphologic future analogue to Svalbard.

In the recent IPA action group on rock glaciers' categorization, the term 'transitional' has been introduced for partly dynamic rock glaciers, those that are not active anymore, but not fully relict either. Exactly which processes the measured slope dynamics represent are difficult to interpret, most likely there is a combination of different processes; seasonal creep, rockslides, rock fall, snow avalanches, etc. This highlights the value of combining different methods in order to interpret the observations. It also shows that the traditional terminology of "relict" and "intact" landforms is not always sufficient (IPA, 2020).

**6 Conclusions**

From this study, the following main conclusions can be drawn:

- Coastal rock glaciers in northernmost Finnmark are widespread, and entirely conditioned by the bedrock type, with the major occurrence in the quartzite belts in the area.
- These rock glaciers may have formed after the early deglaciation in Late-Pleistocene. Rock glacier activity has probably varied between stages with variable movement rates at several time periods in accordance with the Holocene climate fluctuations. In Ivarsfjord rock glacier, and in several of the relict rock glacier systems in the region, we find upper lobes with currently higher movement rates than the well-developed lower parts. These observations could indicate partly active rock glaciers, or younger generations of rock glaciers developed on top of the relict ones in colder time periods of the Holocene.
- The rock glacier observations from Hopsfjorden can be interpreted as landforms in transition from an active to a relict state in response to climate changes. The landforms that were initially interpreted as either intact or relict from aerial photos, are in fact *not* significantly different in terms of movement rates, exposition or climate. Our combined observations suggest that the location and visual appearance of the rock glaciers in the Hopsfjorden area is mostly dependent on the local bedrock and topography. However, the observed below zero summer temperatures in parts of the Ivarsfjorden rock glacier suggests that minor ice bodies in the landforms are still present.
- Our study finds relatively complex systems of rock glaciers, talus, landslides, and scree in close vicinity with variable creep rates depending on lithology and ground thermal state. These different landforms have similar morphology. This illustrates 1) the need of combining several methods when characterizing mountain permafrost landforms, and 2) the drawback of the traditional terminology to describe rock glacier activity state. It may be more accurate to address these systems as complex creeping systems that exist in various states of transition between a fully active rock glacier to a fully stabilized relict landform.
- Our conclusions could not have been drawn without the valuable combination of different remote sensing and *in situ* methods to provide a comprehensive understanding of such complex and dynamic systems. We believe that this study demonstrates the benefits that come with extensive field investigations combined with different remote sensing techniques.

**Data availability**

The regional InSAR data analysis of the rock glaciers in this study is based on the ESA Climate Change Initiative (CCI) Permafrost CCN2 project (4000123681/18/I-NB), and the data are freely available at https://unifr.maps.arcgis.com/apps/instant/portfolio/index.html?appid=99cf50eb91c245d1b171a4a842d8ef0e. All other data are available opon request.

**Author contribution**

KSL has initiated the study, and written the drafts of the manuscript, supervised and participated in the field work and analysed the thermal information presented in the study. BE has revised and commented the manuscript and carried out/analysed the ERT measurements. LR has processed the InSAR data and analysed kinematics at the regional and landform scale. TE has carried out the drone SfM and TLS data. CH performed and analysed the RST. The geomorphological maps and the analysis of the aerial photos are based on two MSc theses (Vetle Aune, 2018, and Gaute Slinde, 2020). All authors commented on the

final versions of the manuscript.

**Competing interests**

None.

**Acknowledgements**

This study was partly funded by the project 'CryoWALL – Permafrost slopes in Norway' (243784/CLE) funded by the Research Council of Norway (RCN) and mostly by the Department of Geosciences, University of Oslo. The geomorphological mapping was part of the MSc thesis by Gaute Slinde (http://urn.nb.no/URN:NBN:no-87852), and the analysis of orthophoto changes were provided by the MSc thesis by Vetle Aune (http://urn.nb.no/URN:NBN:no-66987). Rouyet's PhD is funded by

the Research Council of Norway (FrostInSAR project nr. 263005). The InSAR-based kinematic inventory and update of the rock glacier inventory has been supported by the ESA Climate Change Initiative (CCI) Permafrost CCN2 project (4000123681/18/I-NB) (https://climate.esa.int/en/projects/permafrost/). Sentinel-1 scenes were provided by the EU Copernicus data service (2015–2020). We acknowledge the International Permafrost Association for its support to the Action Group 'Rock glacier inventories and kinematics', which is defining international recommendations towards standardized rock

glacier inventories and time series of permafrost creep rate as climate change indicator (https://www3.unifr.ch/geo/geomorphology/en/research/ipa-action-group-rock-glacier/). Valuable help in field was provided by Kristian Fagernes, Harald Hestad, Maria Peters, Léo Martin, Coline Mollaret, Guy Ramsden, and others. The refraction seismics were analysed at the University of Fribourg, Switzerland.

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

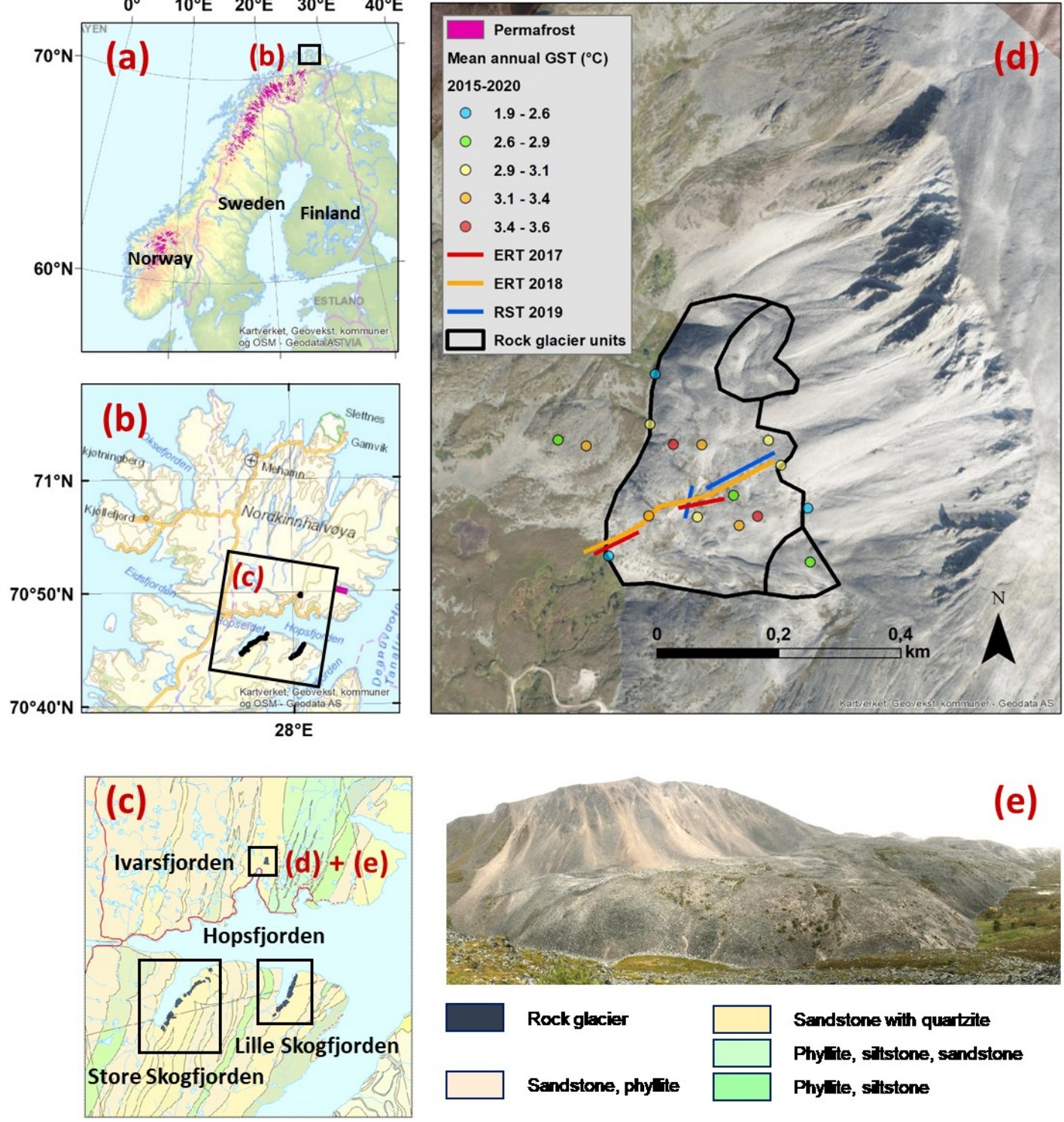


**Figure 1: A: Overview map of Norway, Sweden and Finland (© ESRI and OpenStreetMap contributers), with our study area indicated by the black square B. Probable permafrost presence as modelled in Gisnås et al. (2017) is shown in pink. B: Closer view**

of Nordkinn peninsula, with the location of the inlet map C (© Norwegian Mapping Authority). Rock glacier units are marked in black. C: Geological map of the Hopsfjorden area (©Norwegian Geological Survey). The black boxes are similar to the geomorphological maps in Fig. 2. D: Ivarsfjorden rock glacier, with mean annual ground surface temperatures (GST; 2015-2020). The lines on the rock glacier are profiles of Electrical Resistivity Tomography (ERT) from two years (2017; red, and 2018; orange) and Refraction Seismic Tomography (RST) from 2019 (blue). E: Picture of the Ivarsfjorden rock glacier, taken towards east.

## Legend

- ᴹ Till
- ⏜ Hummocks
- ⟐ Polygons
- ⌒ Solifluction lobe
- △ Boulder-rich surface
- ✳ Patterned ground (frost-sorted landforms)
- ▫ Debris flow depost
- ᵁ Beach ridge
- ᴱ Fluvial deposit
- ˢᵖ Rock fall
- ↣ Relict drainage parth
- ↠ Gully
- ↣ Drainage path, active
- ↣ Drainage path, relict
- ┬┬┬ Terrace edge
- ▪▪▪▪ Beach ridge
- ------ Avalanche deposit
- ⟶ Debris flow track
- – – – Distinct avalanche path
- ▪—▪— Scarp
- ⎯⎯ Rock glacier front edge

| | |
|---|---|
| 🟩 | Till |
| 🟦 | Marine deposit, unspecified |
| 🟨 | Fluvial deposit |
| 🟪 | Weathered material |
| 🟪 | Mass-movement material |
| 🟥 | Rock glacier, fresh surface |
| 🟫 | Rock glacier, relict surface |
| 🟪 | Rock fall deposit |
| 🟪 | Snow avalanche deposit |
| ▨ | Debris flow deposit |
| 🟫 | Solifluction material |
| 🟫 | Stone-rich solifluction material |
| 🟨 | Exposed bedrock |
| 🟨 | Exposed bedrock, thin sediment cover |

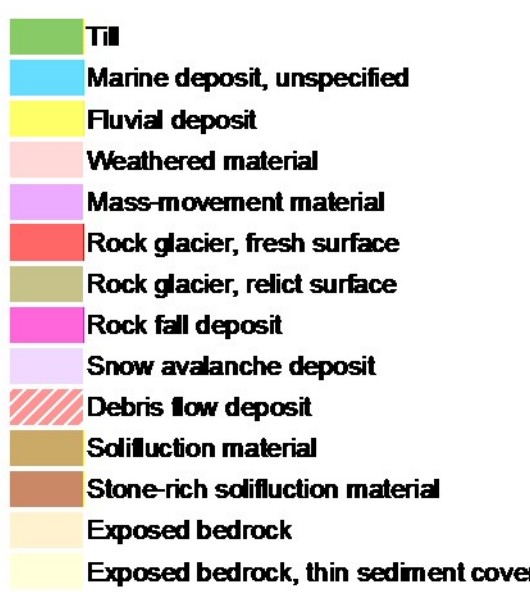

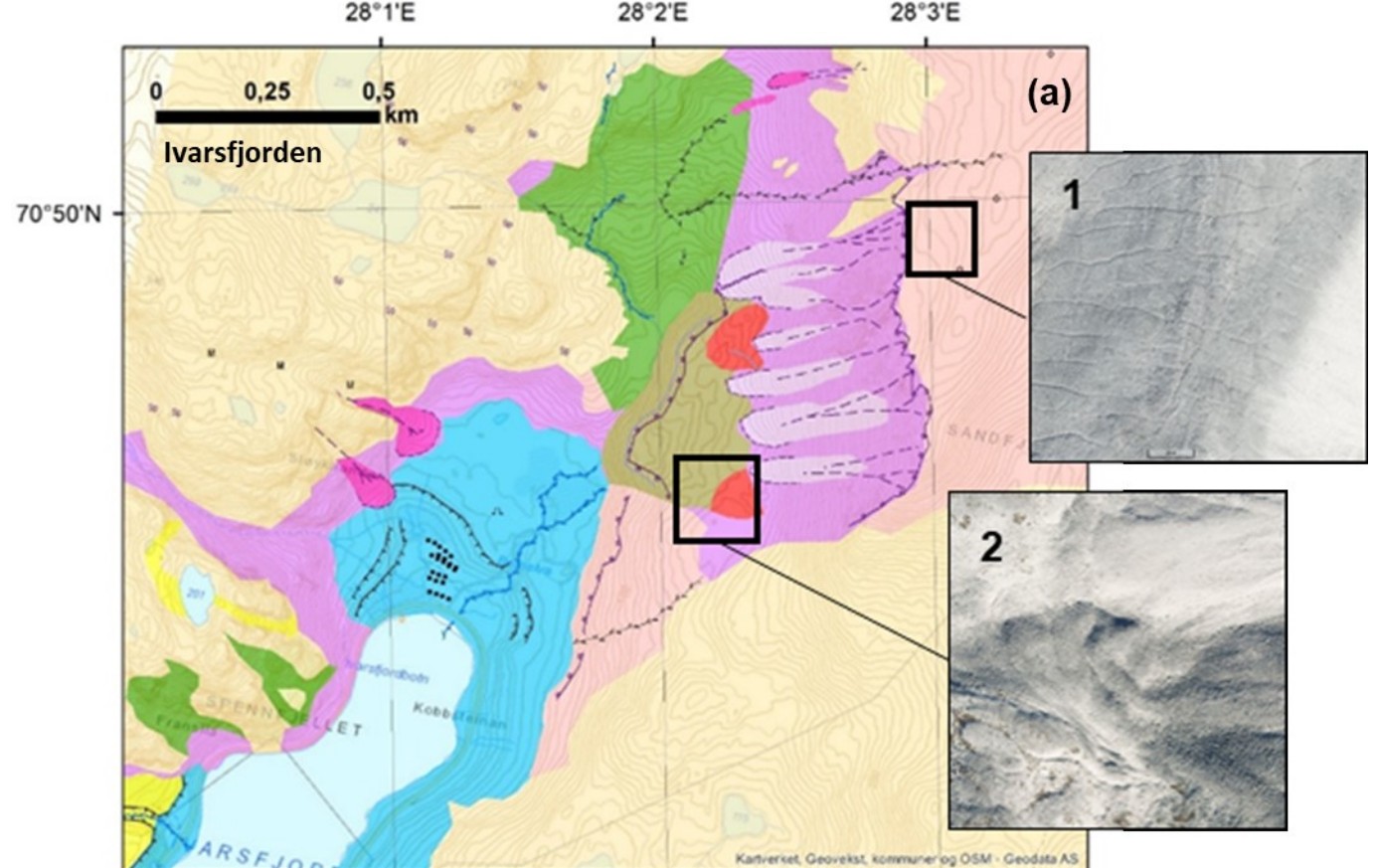


**(b)**

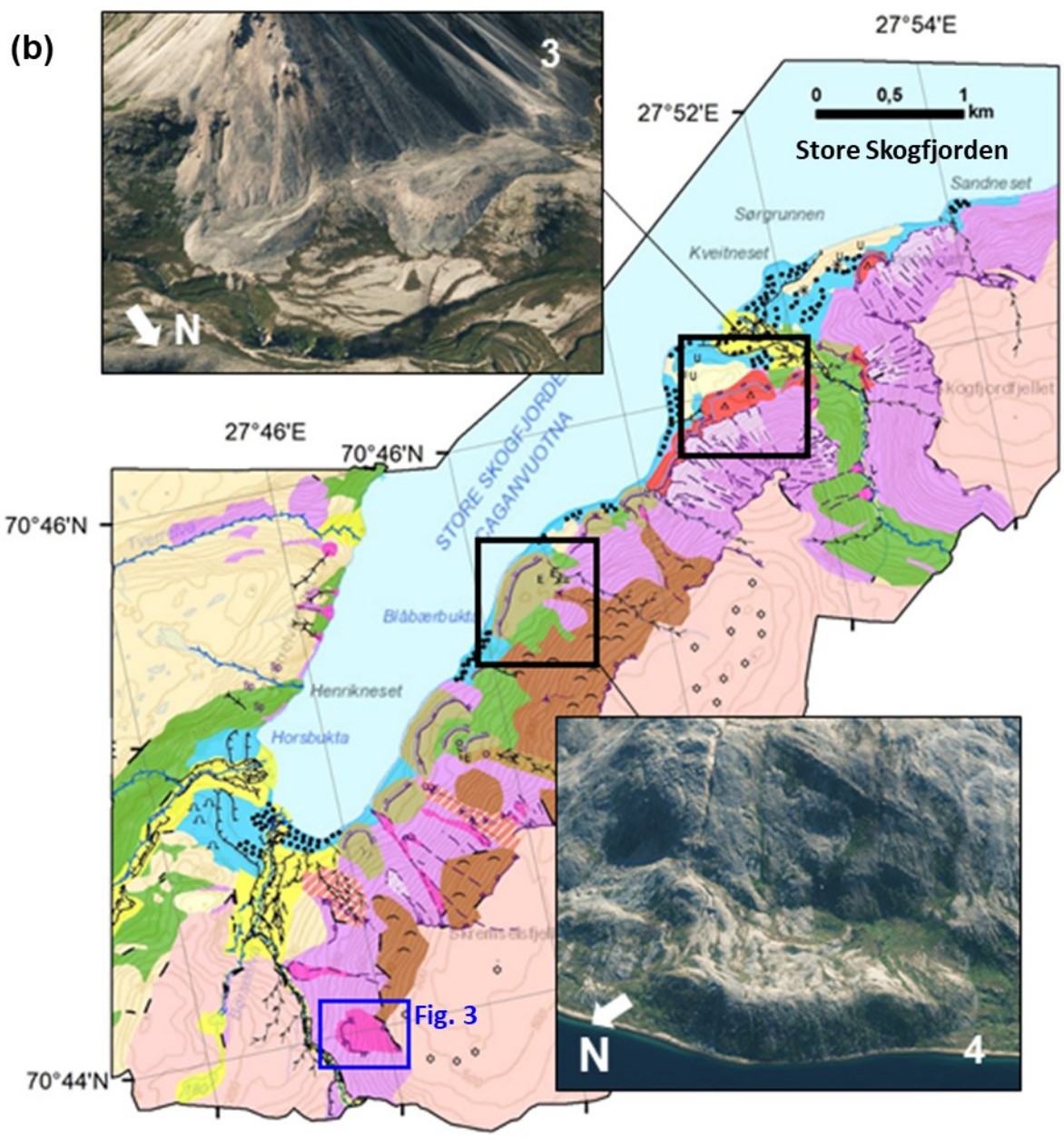

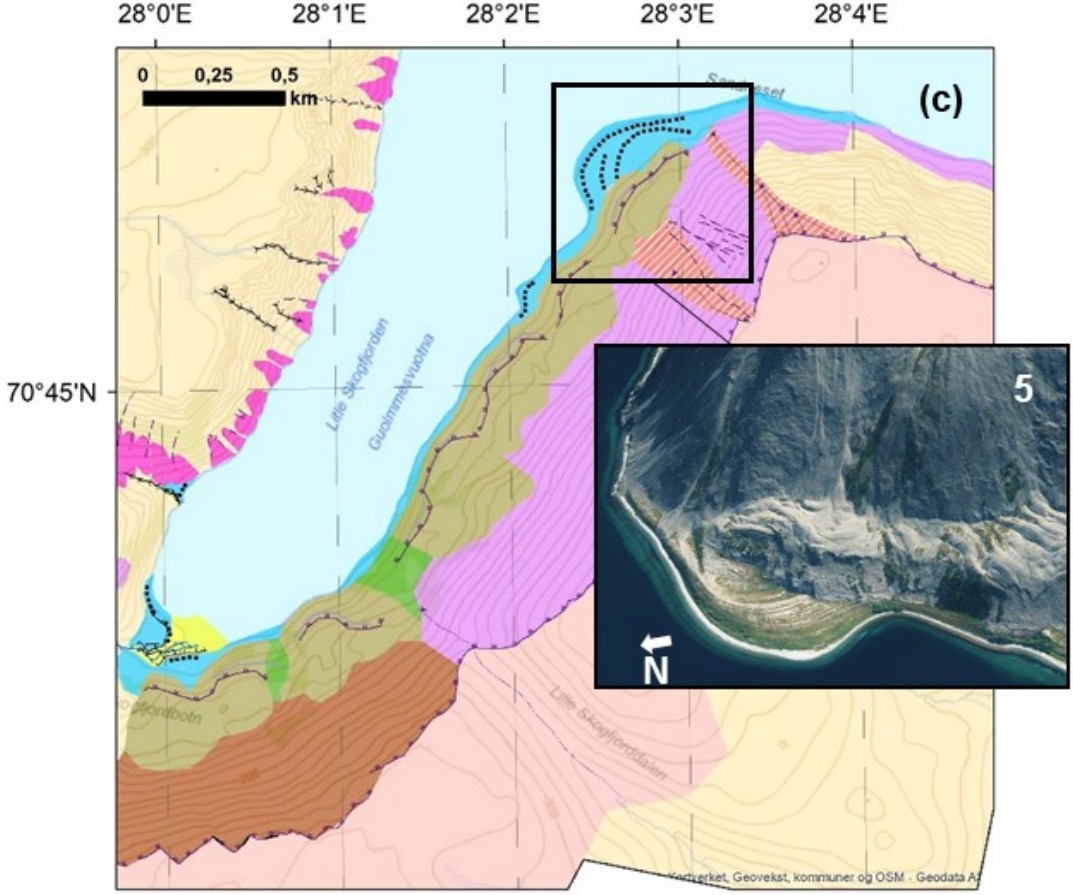


**Figure 2: Geomorphological maps of (a) Ivarsfjorden (including the closely investigated Ivarsfjorden rock glacier), (b) Store Skogfjorden, and (c) Lille Skogfjorden. The inset images show from the top: 1. polygonal network pattern on the Sandfjellet plateau, 2. the currently most active part of Ivarsfjorden rock glacier, 3. intact rock glacier north in Store Skogfjorden, with raised shore lines, 4. relict rock glacier in mid-Store Skogfjorden, and 5. intact rock glacier crossing raised shorelines north in Lille Skogfjorden.**

**The blue square southwest in fig. 2b indicates the location of the rockslide shown in Fig. 3. All topographical background maps are the owned by the Norwegian Mapping Authority.**

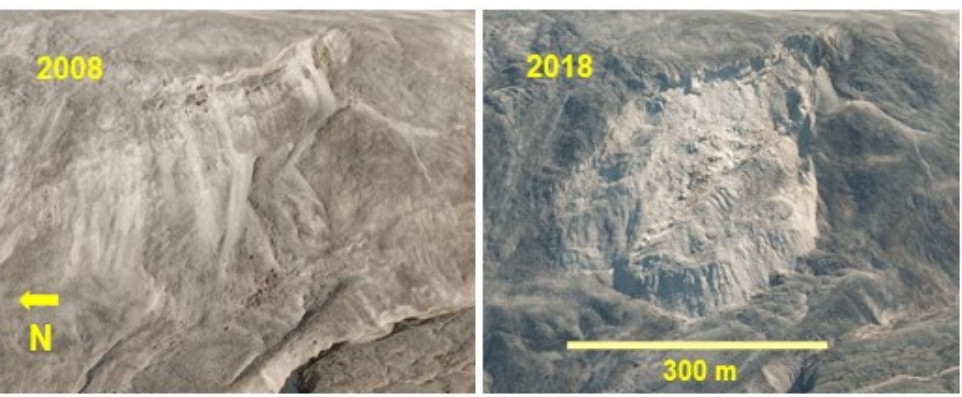

**Figure 3: Rockslide south of Store Skogfjorden as it appears in aerial photos prior to the event (2008) and after it occurred (2018). Photos are retrieved from www.norgeibilder.no (3D version), owned by the Norwegian Mapping Authority. The 2008 photo, part of** 
**the Kongsfjorden campaign, was taken on the 19th August 2008, while the 2018 photo, part of the Finnmark campaign, was taken on the 18th July 2018. The rockslide is about 300 m wide, and the upper scar is 15-20 m tall. The location is shown in Fig. 2B.**

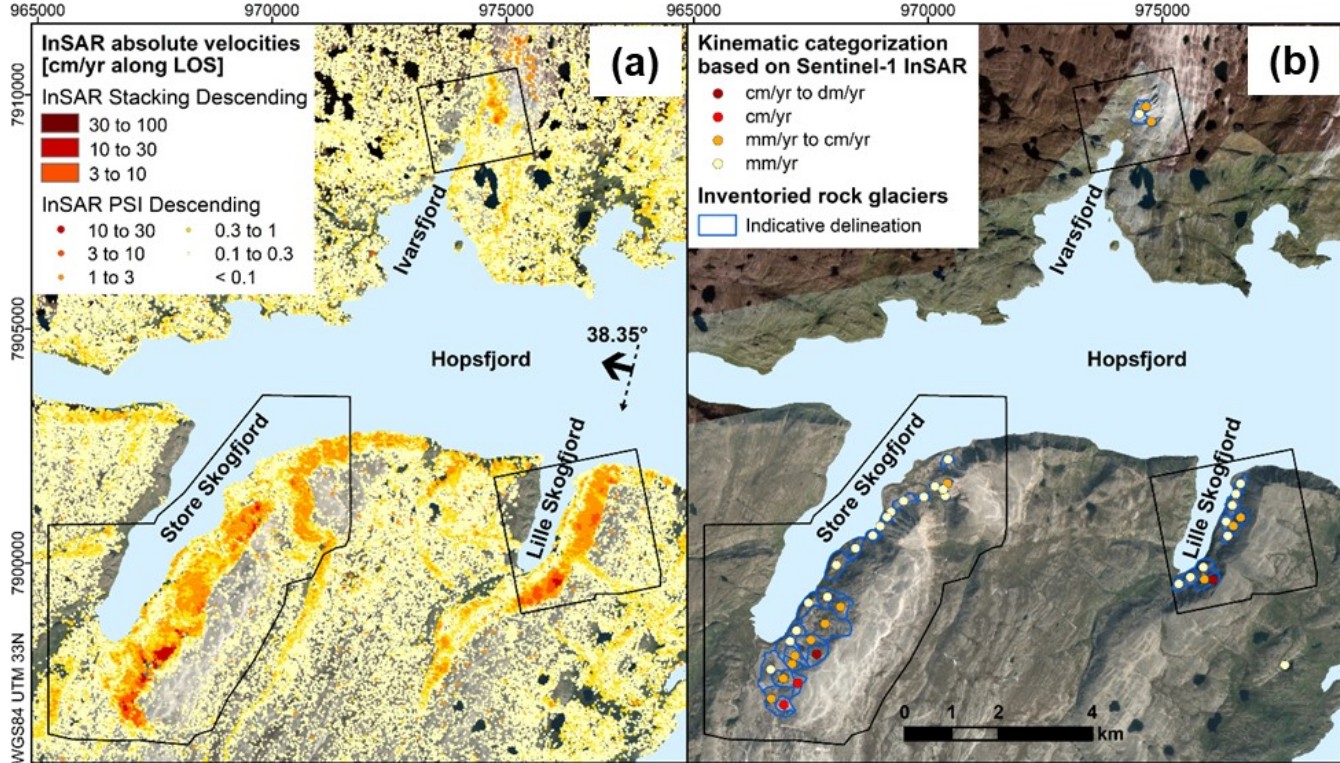

**Figure 4: (a)** Composite InSAR map based on classified mean annual ground surface velocities from snow-free Sentinel-1 satellite images (June-October) between 2015 and 2020 (descending geometry). The map combines results processed with InSAR Stacking method (Rouyet et al., 2021) and Persistent Scatterers Interferometry (PSI) (NGU, 2020; Dehls et al. 2019), two methods that have complementary detection capabilities and therefore cover different velocity ranges. The black polygons show the extents of the geomorphological maps from fig. 2. The black arrow shows the direction of the flying satellite (dashed line) and the corresponding radar line-of-sight (LOS) towards WNW. The label above the LOS arrow indicates the angle between the vertical and the LOS. **(b)** Indicative delineation of rock glacier units and associated kinematic attribute (order of magnitude of the creep rate) based on Sentinel-1 InSAR velocity from (b). The backgroundimagery are from "Norge i bilder" © Norwegian Mapping Authority, Projects: Nordkinnhalvøya 2012, Gamvik Lebesby Tana 2016, and Finnmark 2018. Retrieved from https://www.norgeibilder.no/

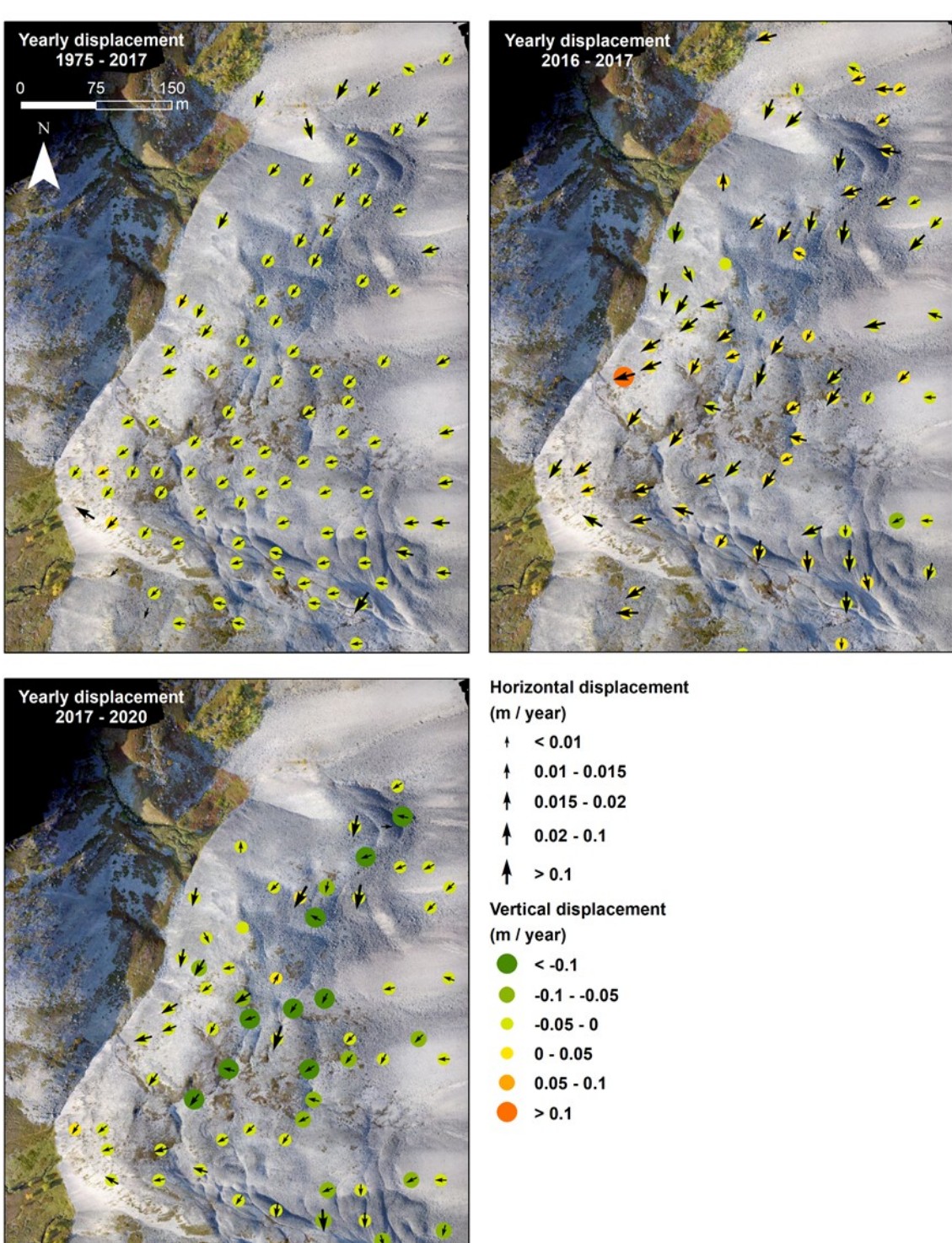


**Figure 5: Yearly displacement of Ivarsfjorden rock glacier between in the time periods 1975-2017 (top left), 2016-2017 (top right), and 2017-2020 (bottom left). The arrows show length and direction of horizontal movement, while the circles show vertical changes. Negative values means lowering of the ground, positive values means higher ground in the time periods. All numbers are given in meters per year.**


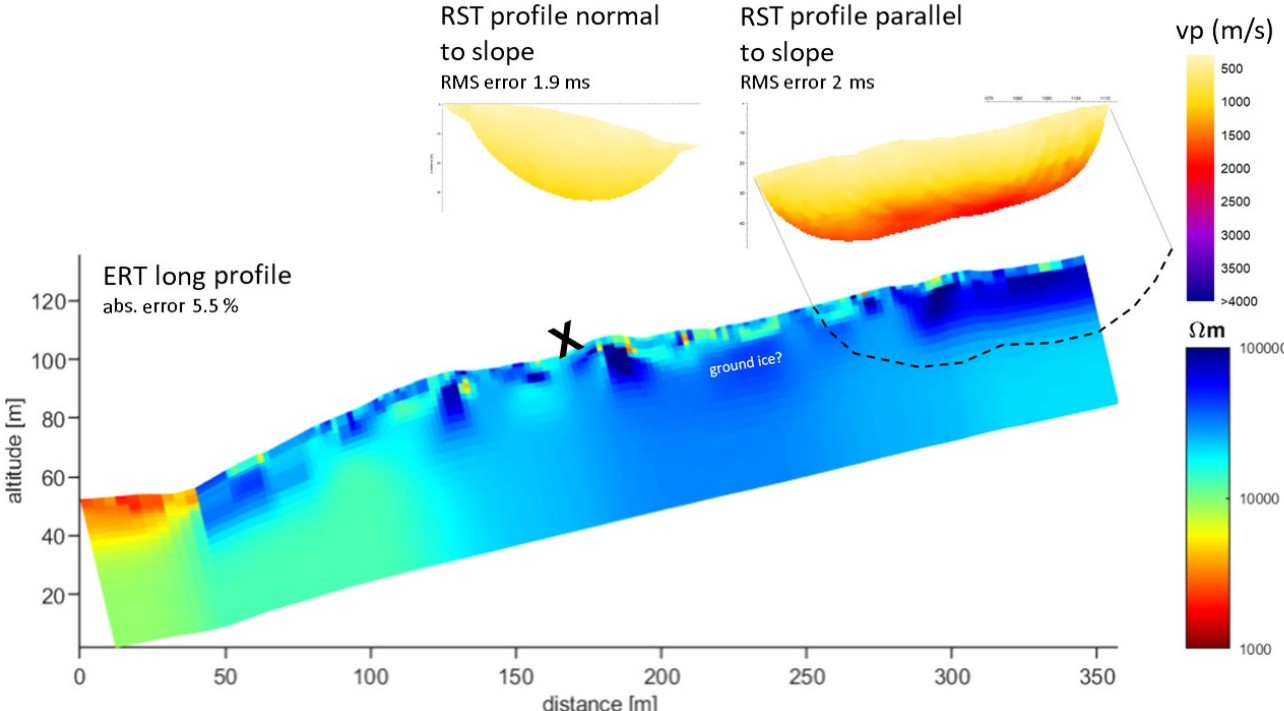

**Figure 6: The long Electrical Resistivity Tomography (ERT) profile retrieved in 2018, and two Refraction Seismic Tomography (RST) profiles retrieved in 2019 (all locations in fig. 1). The X marks where the RST profile normal to slope crosses the ERT profile. Blue colours in the ERT profile mean high resistivity (above 10 000 Ωm), while yellow and red colours in the RST profiles means low wave propagation (below 2000 m s⁻¹).**

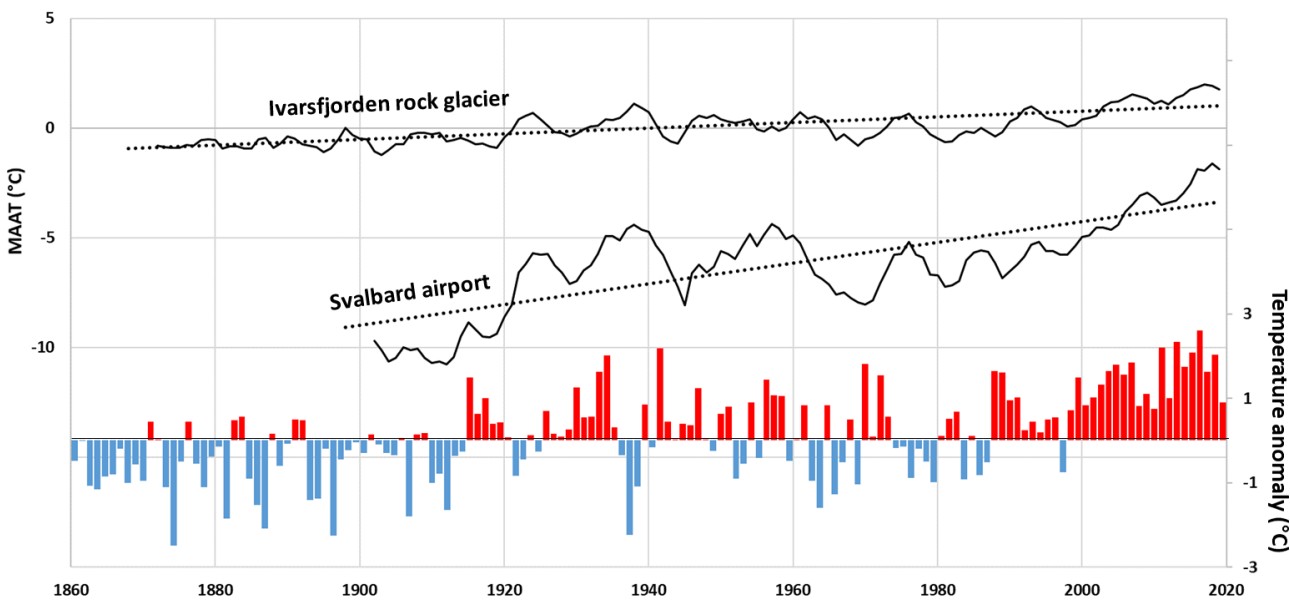

**Figure 7: Mean annual air temperature of Ivarsfjorden rock glacier (combined by SeNorge data from 1957 to present, and by extrapolating the data series based on Vardø radio temperature observations starting in 1868), and the composite temperature records from Svalbard airport (Nordli et al., 2014) updated to 2020. Both series are shown as the running average of 5 years (full line) and the linear trendlines (dotted lines). The lower graph is a temperature anomaly plot of yearly temperatures at Ivarsfjorden rock glacier compared to the 1961-90 normal period.**

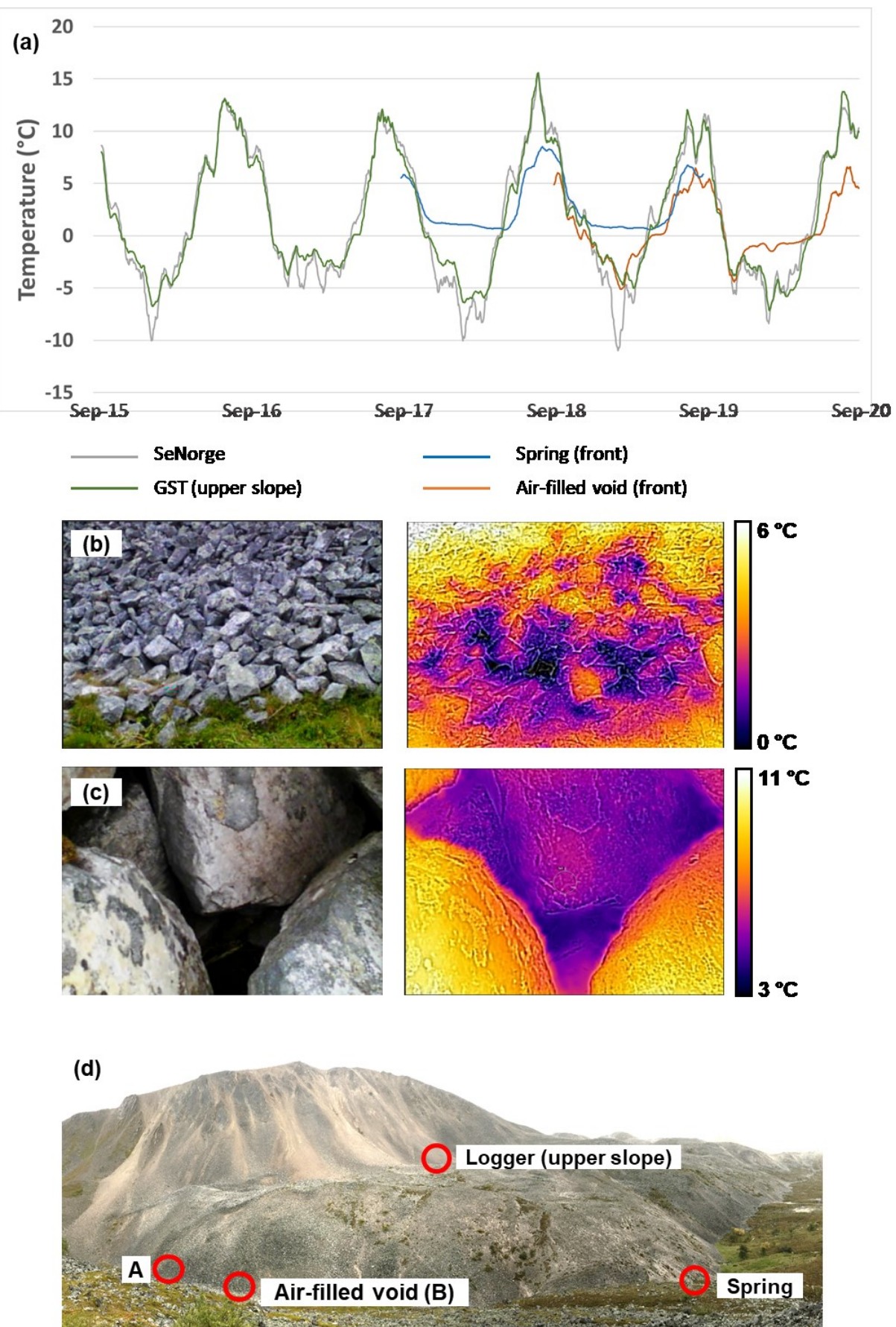

Figure 8: (a): Air and ground surface temperature over the 5 years of monitoring. The graph shows daily temperatures smoothed over 20 days; grey: SeNorge air temperature; green: ground surface temperature (GST) close to the top of the rock glacier; blue: temperature of water from spring escaping from the rock glacier front; orange: open blocks in the rock glacier front (positions indicated in image (d). Images (b) and (c) show optical (left) and thermal (right) image of the rock glacier front. Positions can be found in image (d). Image (b) is also the location of the temperature measurements in orange above. Both images (b) and (c) are taken on 10th Sept. 2018. This day had air temperatures above 20 °C.


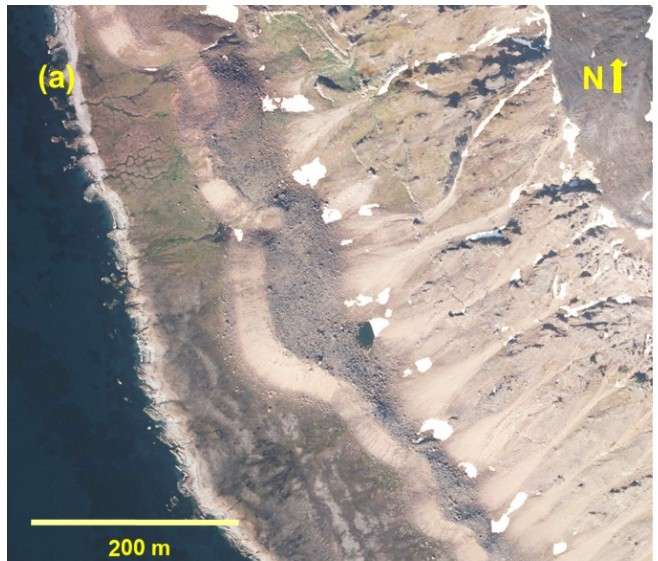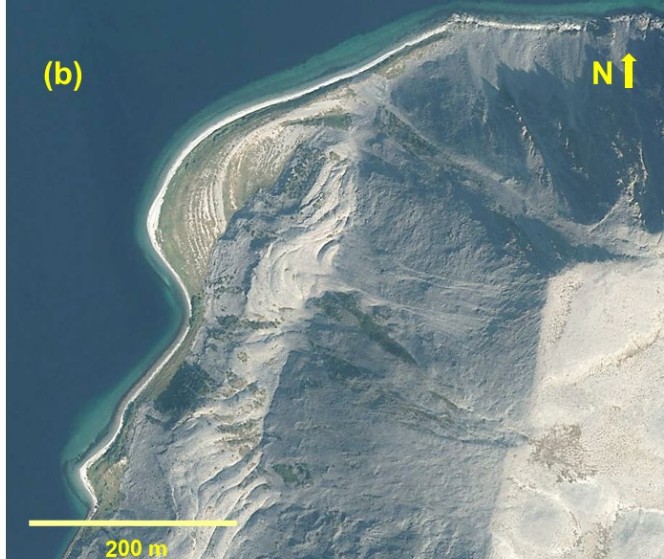

**Figure 9: Comparison of landforms in (a) Svalbard (Prins Karls Forland; Western Spitsbergen; © Norwegian Polar Institute) and (b) Lille Skogfjorden, Finnmark (© Norwegian Mapping Authorities). In both cases the rock glaciers creep from talus slopes onto the strandflat. Especially in (b) the relict shorelines are visible.**

**Table 1: Estimated height accuracies of the DEMs used in this study. The year refers to the time the images were collected to produce the DEMs.**

| Year | Accuracy (cm) | |
|------|---------------|---|
| 1975 | 12 | Historical aerial photos |
| 1982 | 41 | |
| 1992 | 38 | |
| 2016 | 0.1 | Drone photos |
| 2017 | 0.3 | |
| 2017-2020 | 2-4 | TLS |

**Table 2: Vertical and horizontal displacement rates (cm/year) of different pairs of DEMs and orthophotos from Ivarsfjorden rock glacier. TLS: Terrestrial Laser Scans.**

| Period | Vertical displacement rates | | Horizontal displacement rates | | |
|--------|-----------------------------|---|-------------------------------|---|---|
| | Range (cm yr$^{-1}$) | Mean (cm yr$^{-1}$) | Range (cm yr$^{-1}$) | Mean (cm yr$^{-1}$) | |
| **1975-1982** | -30 – 10 | -10 | 0 – 2 | 1 | Historical aerial photos |
| **1975-1992** | -10 – 0 | -5 | 0 – 5 | 1 | |

| | | | | | |
|---|---|---|---|---|---|
| **1975-2017** | -10 – 0 | -7 | 0 – 3 | 1 | |
| **1982-1992** | -10 – 20 | -5 | 0 – 6 | 3 | |
| **1982-2017** | -7 – 5 | -2 | 0 – 2 | 1 | |
| **1992-2017** | -10 – 10 | -3 | 0 – 2 | 1 | |
| **2016-2017** | -10 – 5 | 0 | 0 – 3 | 2 | Drone photos |
| **2017-2020** | -5 – 2 | -1 | 0 – 3 | 2 | TLS |

