# Peer review of "Transitional rock glaciers at sea-level in Northern Norway"

_Earth Surface Dynamics, 2022_

## Referee Comment (RC2)

[revised manuscript text omitted]

---

## Author Comment (AC1)

**Response to referee 1**

**Our comments in green.**

**General feedback**

I enjoyed reading this interesting manuscript. The complex behavior of degrading rock glaciers in response to climate change can only be decoded in a multi-methodological approach, as you nicely show. The final decision as to whether the landforms are active, inactive/transitional or relict is not an easy one – especially when some results are contradicting. But, you show that it is still important.

Dear referee,

Thank you so much for finding our study interesting, and for taking your time to both read and carefully give us feedback on our manuscript. We are grateful for your contribution to making our paper much better than the first draft you went through.

Overall, the manuscript is well written and structured. While the methods and results part is totally fine, I have some suggestions to improve the reasoning in the paper (from your motivation to your conclusion). More background on the relation of permafrost occurrence and rock glacier activity should be given in the introduction to better indicate the motivation of this study (you provide this in the discussion, chapter 5.1, but part of the information should be provided earlier). What is the link to the inventory you did in 2011?

We have made several changes to the introduction section. We have added a few sentences, such as (L40-ff):

"Rock glaciers form when ice-cemented ground starts to creep because of the ice's plasticity and gravity (King, 1986; Delaloye and Lambiel, 2005; Farbrot et al., 2007; Haeberli, 1985; Berthling, 2011). Since rock glaciers are visual expressions of permafrost, the distribution of both active, inactive and relict landforms is indicative of certain climate types at present, in the recent past, and in the more distant past. Therefore, in 2011, a Norwegian rock glacier inventory of 307 landforms was published (Lilleøren and Etzelmüller, 2011), mainly mapped based on the digital aerial photos available at that time. This mapping was based on visual interpretation of the landforms, i.e. convex shape, clear front slope, signs of surface creep or thermokarst, situated downslope of talus or other debris deposits, and included activity status. The quite few examples of mapped rock glaciers close to sea level are situated in areas that were deglaciated before the Younger Dryas (YD), and where climatic conditions favoured permafrost development outside the Weichselian ice margin during the deglaciation (Andersen, 1981; Sollid et al., 1973). These rock glaciers originate exclusively from talus slopes. They sometimes cross raised shorelines, and they vary in size and shape from small lobate landforms to more developed tongue-shaped or lobate landforms, and were always interpreted as being relict.", but we mostly tried to clarify the former text. We hope this part is clearer now.

You use the term "transitional" or mention "active to relict stages" (e.g. l. 27, l. 430) in reference to the IPA WG. These terms are related to the kinematics, not including the ice content (as e.g. the term "inactive" does, see Barsch 1996). But, you also apply geophysical soundings to find indications for ground ice occurrence. This of course is important to link back to permafrost distribution. Here, some more background will support your argumentation.

We have tried to adjust the text so that 'transitional' is only used where it clearly refers to the dynamics of the rock glacier (e.g. l., 59, 459), while 'intact' is always used in a geomorphological context (e.g. l.60, 212 ff).

The structure of the paper (especially chapter 4 and 5) is very individual and interesting. The different sections could be linked better in some parts (introduction, discussion, and conclusion). For example, the motivation to compare the rock glaciers in Northern Norway to those in Svalbard (analogues) could be mentioned before.

Thank you again for useful comments and suggestions, we have tried to clarify these parts, and also tried to better show the connection between the different sections, e.g. L74 ff: "We specifically investigate the dynamic state of Ivarsfjord rock glacier and the other rock glaciers in the region. We interpret the formation period and how they have developed over time, and we suggest this area to be an interesting analogue to the development of Svalbard rock glaciers in a warmer future."

Individual edits are listed below.

**Introduction:**

l. 34: the name of the author is Vonder Mühll (please correct) and reference is missing in reference list.

Corrected.

l. 41: add more information on geomorphological characteristics of the landforms, as important for the mapping. Any specific characteristics for rock glaciers at sea level (e.g. short lobes)?

We have added two sentences at l. 43 and l. 47, that hopefully clarifies this point.

l. 52: add link to official document of IPA working group (or the website)

Done.

l. 55: definition of an intact rock glacier; so, different to the one given by Barsch?

Thanks for pointing this out, this should not be stated as a "definition", but an interpretation. We have changed this to not contradict Barsch's definition (l. 58-60).

l. 56: regional analysis; are this selected areas with rock glaciers at sea level? And if, why this selection?

This sentence has been changed to: "While the coastal rock glaciers in other parts of Norway are interpreted as relict based on their visual appearance, partly overgrown with vegetation and with relatively smooth topography, the rock glacier clusters in northernmost Finnmark appear more active with steep fronts, ridges and furrows."

l. 58: where is this information on activity coming from?

This is only based on visual interpretations of the geomorphology from aerial photos. We tried to clarify this sentence (l. 63-65) and it now reads "While the coastal rock glaciers in other parts of Norway are interpreted as relict based on their visual appearance, partly overgrown with vegetation and with relatively smooth topography, the rock glacier clusters in northernmost Finnmark appear more active with steep fronts, ridges and furrows.."

The last paragraph of the introduction and the aims of this study, respectively, could be linked to the titles/questions given in the discussion.

We have tried to adjust this part of the introduction according to your suggestions. It now reads: "We specifically investigate the dynamic state of Ivarsfjord rock glacier and the other rock glaciers in the region. We interpret the formation period and how they have developed over time, and we suggest this area to be an interesting analogue to the development of Svalbard rock glaciers in a warmer future. Thus, this study provides new insights about former and present permafrost distribution in this sub-arctic environment, and how these rock glaciers near sea-level transition in response to a changing climate, both post-deglaciation and more recently."

**Setting:**

l. 82: … the northern areas… "were" (not was)

Corrected

l. 99 ff: in this paragraph may be mention that the relict rock glaciers are indicators for the former existence of permafrost (as the frost polygons)

Thank you, this point has been added.

**Methods:**

l. 161/173: complementary geophysical investigations… should be applied in the same conditions (in the same time) as ground conditions may change over time

Yes, it is unfortunate that we were not able to do this at the same time. We have added a sentence about this issue in L189-910: "A weakness in our methodology is that we were not able to apply the ERT and the RST measurements at the same time, although we are aware that ground conditions may change over time."

Method/potential of the thermal camera is not mentioned. Provide some details about the accuracy and give references.

We have added the methods, specifications and accuracy of the thermal camera.

**Results:**

l. 191: "… fig c))…" close the second bracket

done.

l. 229: rather rock glacier units instead of "rock units"

done

l. 245: "Fig 5c", A-C are not labelled in Fig. 5

corrected.

l. 279: "until now", I suggest to give the year (2019)

changed

**Discussion:**

l. 289: give references to permafrost models

done

l. 320: "…snow variations" add also "variations in material properties" (block size, etc.)

done

l. 336: here is would be helpful to interpret the vertical and horizontal deformations independently to differentiate between processes. With the CIAS program you match single

blocks at the surface; if you see a consistent flow field, the blocks are for sure not moved by an avalanche or rock fall but by solifluction or permafrost creep.

Thank you for this input. We think we see both processes at work in our study site. Consistent and slowly moving patterns are from solifluction or other frost creep as you point out, while occasional fast moving rock masses are something else. A more detailed investigation of the vertical/horizontal movement vectors would indeed be valuable but have not be performed in this study because it was out of the scope of the paper.

l. 375: Did you map several lobes on top of each other, this could be a morphological indicator for reactivation phases?

Yes, both you and the other reviewer address this point, and we have tried to adjust the text accordingly. The mapping did reveal more active lobes at the upper areas of several rock glaciers, which in fact could indicate reactivation.

l. 411: "… are probably…?.. because of" do you mean "are low because of…"

Yes, corrected.

l. 420: write "IPA" not "Ipa"

Corrected in multiple references.

**Conclusion:**

In this chapter, you could mention the valuable combination of different remote sensing and in-situ methods to better understand dynamic systems. Without the extensive field work or the available remote sensing data on the other hand, you would not be able to draw your conclusions.

Thank you, we have added some text based on this feedback, and the last bullet point of the conclusions now reads (L487 ff): "Our conclusions could not have been drawn without the valuable combination of different remote sensing and *in situ* methods to provide a comprehensive understanding of such complex and dynamic systems. We believe that this study demonstrates the benefits that come with extensive field investigations combined with different remote sensing techniques."

l. 436: this point of the problematic complex geomorphological systems of rock glaciers, talus, landslides, and scree in close vicinity is not really elaborated within the paper and should be detailed e.g. in chapter 4.1

We have addressed this now in ch. 4.1, L222 ff: The slope systems are generally complex including a wide range of processes, from solifluction to rock falls feeding into talus slopes, and landslides reaching the shoreline. Several of the processes are superimposed, like solifluction on an old landslide, or frost creeping talus cones.

Figure 1:

Section A: country names are not readable

Changed

Are there different nuances of pink in the permafrost distribution map (it looks like lighter and darker areas)?

Changed

Section C: indicate D + E for Ivarsfjorden

Changed

Figure 2:

Add the area names to the different map sections

Added

Maps and Photos are too small.

These have been enlarged.

Symbols and lines are hard to differentiate.

This is hopefully also better with larger maps.

Figure 4

A: Legend does not fit to the colour code in the figure. There are no circle symbols (PSI) and field colors (stacking) does not cover the entire range from light yellow to dark red.

The current figure (the legend and color code) is correct. The legend is split into two different ranges because the map is an overlap of two products processed with different InSAR methods (Stacking and PSI) leading to different detection capabilities: Stacking is not tailored for low velocities but allows for documenting velocities up to 30-100 cm/yr, while PSI is not tailored for high velocities but well suited for mm-cm/yr velocities. So it is natural

that the two data sources do not cover the entire range – that is basically the point of including both. We have clarified this point at l. 137-143 and in the figure caption.

Figure 5

A, B, C are not indicated in the Figure (as given in l. 245)

We have removed the reference the fig 5C in the text, since we considered it not to be crucial.

Figure 9

Image B is partly very dark, please improve.

We have brightened fig 9B.

Table 1

Is the accuracy really given in cm? This should be rather meters (e.g. TLS 2-4 cm accuracy)

Yes, we have changed the numbers to cm now.

What is the accuracy of the orthophotos?

The accuracy of the orthophotos is between 12 and 41 cm.

Table 2

Very small displacements and mainly in the area of the accuracy. Can you highlight the numbers that are not in the range of the accuracy?

We disregarded all the changes below 50 cm for the DEMs made from orthophotos, meaning that the listed displacements should be outside the accuracy level. We mention this in line 255 in the text: "We estimated the lowest accuracy for any of the DEMs derived from the orthophotos to ±0.41 m, thus all difference values below 0.5 m were discarded." The TLS and drone photo DEMs have accuracies of less than 10 cm.

**Response to referee 2**

**Our comments in green.**

This is a very interesting paper showing the complexity in behavior of rock glaciers appearing relict using cutting edge technology and methods like InSAR analysis, high resolution photogrammetry and topography measurements using UAVs and TLS, geophysical methods (ERT and SRT), thermal monitoring using miniature data loggers and IR camera and long term climatic analysis.

1) It interprets the rock glacier behavior in light of the new IPA action group Rock glacier inventories and kinematics and it shows that rock glaciers can register slow movement in spite of very little or even no ice underground. The authors show that this might by determined by other processes like solifluction.
   No other paper dealt so far with rock glacier activity in spite of permafrost missing. This paper demonstrates this contradicting situation and suggests that rock glacier movement can occur by other processes than permafrost creep. This is completely new and has not been regulated by IPA action group. However, IPA group indicates that activity of a rock glacier is related to permafrost creep: a moving  area  (...) has to represent the downslope movement rate of the rock glacier (permafrost creep) in the area of concern." So, in the situation of permafrost absence, rock glacier activity in the traditional acceptation should be interpreted with caution and you should discuss this.
2) The paper shows for the first time that rock glaciers can move in spite of MAAT close to 0 and even well above 0 °C. I recommend to discuss more this important finding in relation to other publications showing similar results from marginal permafrost conditions like Serrano et al., 2006 in the Pyrenees and NecÈ™Oiu et al., 2016 in the Southern Carpathians. Especially the latter indicates similar rock glaciers dynamics in order of a few cm/year in apparently relict rock glaciers using similar techniques (InSAR).
3) I tend to doubt about the past activity interpretation of Ivarsfjorden rock glacier. This is seen as active only during the cold phases of Holocene and relict in the rest. The 2°C MAAT increase from LIA to the present is used as an argument. For example, Frauenfelder and Kääb (2000) indicated a similar offset for the relict rock glaciers to be active in the past but that was relative to the year 2000. In my view once depressed topographically and ground voids filled with fines, it cannot reactivate again in spite of ground permanent refreezing. Permafrost oversaturation with ice (necessary for permafrost creep) is not possible any more as the pore space decreased by debris compaction because of interstitial ice melting. I don't know any paper indicating reactivation of the same rock glacier body that was relict when the climate cooled and the possible mechanisms involved. If you know some, please indicate them to support your interpretation. So far, there were documented new lobes generations overriding old rock glacier body (e.g. Amschwand et al., 2021). However, permafrost might be present in the lower talus slope – upper rock glacier contact and thus to be the case of pseudo relict rock glacier as Kellerer-Pirklbauer (2018) indicated.

Please find more punctual comments on the attached reviewd manuscript.

Dear referee,

Thank you so much for your careful considerations of our paper. We are very grateful for you time and effort spent on this review.

You raise several points to consider, and your main issues are discussed below, while your more punctual comments have been addressed directly in the manuscript, and answered to in your comments in the PDF.

1) These important observations and feedback address a blind spot of our work. We have added some lines at the end of section 5.1 (L362 ff) where we discuss this: "This observation also suggests that kinematic data to document surface movement should be used with caution for defining activity and inferring information about the ground ice content of the rock glaciers. We support the definition of the activity as exclusively referring to the efficiency of sediment conveyance (expressed by the surface movement) without any inference on the ice content (RGIK, 2022a). Our investigations generally support the conclusion that a documented creep rate is only one information among others to categorize a rock glacier and should be complemented by other geomorphological criteria. Knowledge of rock glacier ice content is also crucial for a comprehensive characterization of the rock glacier state, but has to be treated independently to its kinematics."

2) In section 5.1 I also added a paragraph on these other studies (L325ff): "However, rock glaciers investigated in other marginal permafrost areas in Europe, such as the Pyrenees or the Carpathians, show both similar and different flow fields as the Ivarsfjord rock glacier. In the Pyrenees, Serrano et al. (2006) observed high surface displacement rates (30-60 cm yr$^{-1}$ along the central flow line) on an active rock glacier in an area with MAAT close to 0 °C. This is in accordance with observations from e.g. the Alps where degrading permafrost and increased temperatures cause the rock glaciers to accelerate (e.g. Kääb et al., 2007). As a contrast, active rock glaciers in Southern Carpathians also in an area of MAAT close to 0 °C, have been observed to move very slowly (typically up to 1.5 cm yr$^{-1}$, interpreted as a consequence of a thin ice-rich deforming layer (Necsoiu et al., 2016). This latter study also reports very similar flow patterns in the areas surrounding the rock glacier compared to the rock glacier itself, suggesting a generally active environment with other periglacial slope processes such as solifluction."

3) Thank you for pointing this out. This truly adds value to the discussion. We think you are completely right, and again you point at a blind spot or an over-interpretation from our side. We have tried to address this issue at several natural parts of the manuscript (L26 (abstract), L242 (results), L257 (results), L412 (discussion), and L469 (conclusions)). We have mapped lobes in the upper parts of the rock glaciers and presented these in the geomorphological maps, but we should of course interpreted these as new generations of rock glaciers.

Direct comments (line numbers refer to the original submitted document, the one commented by the reviewers):

P1, L10: If they are outside of the current models, „should be" should be replaced by „are".

Done.

P1, L13: I dont't think this is to relevant for the abstract here.

"(Ivarsfjorden, Store Skogfjorden, and Lille Skogfjorden)" removed from abstract, as suggested.

P1, L18: I guess you should complete the inventory of the used methods with the IR camera and the climatic analysis.

Added a sub-sentence: "complemented by investigations using an infrared thermal camera, and a multi-decadal climatic analysis."

P1, L21: Please discuss what Tapes shoreline is in the article text.

Thank you, good point. I have changed the wording of the relevant article text ("The most prominent shoreline located by this rock glacier might be the Early to Mid-Holocene shoreline (Tapes) connected to the transgression sea-level rose faster than the vertical uplift of the crust, which according to Sollid *et al.* (1973) is situated ca. 13 m a.s.l."), and also changed the highlighted abstract text from 'Tapes shoreline' to Early to Mid-Holocene shoreline'.

P1, L25: I don't think so because if they are relict now as they are, they were also relict in the warm periods of the Holocene prior to LIA and according to my knowledge, relict rock glaciers cannot be reactivated if the climate cools again. In that situation only a new lobes generation would cover the old rock glacier body but the old one remain dead (relict) and stable. I would suggest that they were active only during Dryas and then moved very slowly by other processes then permafrost creep as you already mentioned they do it in the present.

Yes, very good point. I agree with you, thank you for showing this so clearly to us. This sentence now reads "MAATs below 0 °C 100-150 years ago suggest that new rock glacier lobes may have formed at the end of the Little Ice Age (LIA)."

P2, L34: At the lower altitude of discontinuous permafrost is also influenced by shadowing (microclimate), land cover type (e.g. Deluigi et al., 2017), grain size (Roder and Kneisel, 2012).

Changed, this sentence now reads "The distribution of mountain permafrost is governed by air temperature, snow cover, shadowing, land cover type, and grain sizes; all of these strongly modulated by topography in high-relief settings (Harris and Corte, 1992; Harris and Vonder Mühll, 2001; Rödder and Kneisel, 2012; Deluigi et al., 2017; Gisnås et al., 2017)."

P2, L55: I reccomend you to cite the other document that explain the decision made.

RGIK (2021). Towards standard guidelines for inventorying rock glaciers: baseline concepts (version 4.2). IPA Action Group Rock glacier inventories and kinematics (Ed.), 13 pp.

Changed to: In the study we present here, we have had access to kinematic data of the land surface, and we have therefore used the recent categorization of an International Permafrost Association (IPA) action group on rock glaciers inventories and kinematics. Here, 'active' rock glaciers move more than 0.1 m yr$^{-1}$, 'transitional' between 0.01 and 0.1 m yr$^{-1}$ and 'relict' less than 0.01 m yr$^{-1}$ (RGIK, 2022a).

P2, L54: add „surface".

Based on comments from rev. 1 too, this section now reads:

"In this study, we sometimes refer to landforms as 'intact' landforms. In these cases, we mean the geomorphological appearance of the landforms (Barsch, 1996). However, with one exception, we do not have any evidence of the ice content of the rock glacier interiors."

P2, L58: Can you detail how it will change?

This section has been rephrased to: "If these rock glaciers are active, this would mean that the permafrost distribution in the northernmost coastal areas of Norway are more widespread and situated at lower elevations than what is considered today based on the available models (e.g. Farbrot et al., 2013; Obu et al., 2018; Gisnås et al., 2017)."

P3, L76: I guess you want to cite 1C, the geologic map.

Yes, this has been added.

P3, L78: I don't see this in the reference list.

Sorry, this has been updated with the correct reference.

P3, L87: Please detail what do you mean by that.

Changed to shoreline displacement rates for clarity: "Due to the disappearance of the Barents Sea ice sheet, the shoreline displacement rates (land uplift) in outer part of Finnmark are about three times lower when compared to similar areas in other parts of the

Norwegian coast and the marine limit was reached as early as ca 14.6 cal kyr BP (Romundset et al., 2011)."

P3, L91: The same order like in the first sentence should be maintained, first coast than the interior.

Yes, this has been corrected.

P3, L91 ff: This is confusing because in the results section it seems that the MAAT at the coast is and used to be in fact much higher (-0.6 +1.6). That is crucial for interpreting the rock glacier activity now and in the past. So, please add more details regardin where exactly and on what time interval the MAAT on the coast is of -2 C.

Thank you for noticing. These are numbers that show Tromsø (coast) and Karasjok (interior) at the cold normal period 1961-1990. I have updated the met. data to Slettnes fyr (close to our study area), and Karasjok in the south (interior) for the new normal period 1991-2020. This should make more sense, and I am sorry for not noticing this in the submitted draft. The updated numbers are 2.6 deg for Slettnes and -1.3 deg for Karasjok.

This section now reads: «The climate of this part of northern Finnmark varies between a relatively mild, wet maritime climate at the coast, to a dry continental climate in the interior (Normal period 1991-2020; MAAT from 2.6 °C (coast; Slettnes lighthouse) to -1.3 °C (interior; Karasjok); NCCS, 2021).»

P4, L108: This belongs rather to the study area.

I have moved "In this study we have focused on three areas, Ivarsfjorden (north of Hopsfjorden), Store Skogfjorden and Lille Skogfjorden (both south of Hopsfjorden, fig. 1). All three areas are tributary fjords to the main Hopsfjorden" back to the study area.

P6, L175: The precise type should be indicated. As far as I know, they have 0.5C accuracy. But they can be calibrated in the 0 curtain period when you can really see their error relative to 0 C.

Yes, this has been corrected.

P6, L180: Please indicate in the beginning of the paragraph more clearly what was the purpose of this time series analysis.

Changed to: "To place the investigated rock glacier in a climatic context and to study the historical development of the temperature in the study area, we extracted temperature data

from a gridded climate data set (daily air temperatures and precipitation) available for all of Norway since 1957 at a ground resolution of 1 km."

P7, L191: This should be at study area.

We removed this part from the results.

P8, L245: This figure does not have letters for the subplots.

No, you are right. I will either remove the letter references or change the figure.

P8, L248: Cite the lower plot in fig. 5 here.

Yes, I have added this fig. reference.

P9, L270: 1D.

Corrected.

P9: L277: This is methodology.

I have removed this sentence from the results: "For the decades prior to 1957 we have used the relationship between Vardø radio and SeNorge."

P9, L980: This is in fact figure 8.

Figure reference was corrected.

P11, L320: I would also add shadowing imposed by variable topography and differences in grain size (coarse - colder temp. and fines - higher temperatures).

I have added this, and the sentence reads: "Such variation over short distances are commonly observed in mountainous areas (Gubler et al., 2011), and is attributed to snow variations, topographic shadowing, and variations in material properties including grain sizes (Gisnås et al., 2016)."

P11, L335: Avalanches and rock falls would move the block tens of meters at a time rather than slowly downslope like permafrost creep does. So, solifluction is the only processe possible in my view.

Yes, this was also a comment made by rev 1, and the mention of avalanches and rock falls was removed, and now reads: "This is supported by the facts that most movement is observed in the talus feeding the rock glaciers, and may be attributed to other processes than ice deformation such as solifluction."

P12, L376: 7.

Figure number is changed.

P12, L377: This is not enough to be active though. It should be below -2 or at least below -1C, see e.g. Frauenfelder and Kaeaeb, 2000.

The entire paragraph was changed to: "The rock glaciers may have had several active phases, and colder time periods such as the Neoglaciation and the LIA could have formed several generations of observed rock glacier lobes. The area has warmed by about 2 °C between 1868 and present, and the same between 1957 and present (fig. 7). With an assumed MAAT of 1.6 °C in the 2010-2019 decade, the MAAT could have been just below 0 °C both in the middle of the last century and the end of the 19[th] century. This temperature raise could have triggered a change from an environment where permafrost was sporadically present to an environment with thawing and non-favourable conditions for permafrost. Landslides released prior to the 20[th] century could therefore develop some permafrost, and further some creep, while the presumed permafrost presence is degrading under the current climate. It is however doubtful that completely relict landforms prior to the LIA could have reactivated because of this temperature decrease."

P14, L425: Delete this.

"The existence of" was deleted, and the sentence now reads: "Coastal rock glaciers in northernmost Finnmark are widespread, and entirely conditioned by the bedrock type, with the major occurrence in the quartzite belts in the area."

P14, L428: I think it just varied betwenn relict and pseudorelict throughout the Holocene warm and cold periods respectively.

And: P14, L428: Please disuss more about the past inactive state of the rock glaciers from the studied area because this is only mentioned in the introduction and here at conclusion.

Changed to: "These rock glaciers may have formed after the early deglaciation in Late-Pleistocene. Rock glacier activity has probably varied between stages with variable movement rates at several time periods in accordance with the Holocene climate fluctuations. In Ivarsfjord rock glacier, and in several of the relict rock glacier systems in the region, we find upper lobes with currently higher movement rates than the well-developed lower parts. These observations could indicate partly active rock glaciers, or younger generations of rock glaciers developed on top of the relict ones in colder time periods of the Holocene."

Figure 2:

The names of the fjords (focus stady sites) should be added on each figure.

This has been added.

It is not clear that this is the Ivarsfjorden RG that was analysed in detail. Please indicate that in the figure.

We have added this information to the map ("Ivarsfjorden") and also to the caption.

Please increase the maps and the photo dimensions, and if necessary, add a distinct figure with photos.

We have increased the size of all maps and photos.

Figure 8

Spring is still GST measurement or is the T of water? That would explain the continuous above 0 T.

T of water, corrected.

The upper slope indicate very cold conditions during the winter s end, according to BTS method i indicates permafrost probable at c. -3...-5 C.

The contrast between upper slope and front indicate the potential permafrost in the former and the absence in the latter.

Please discuss this in the paper at the results or discussion section and also please add information in the METHODOLOGY section about HOW did you measure GST (drilled in the rock, how many cm, at what depth etc).

Thank you for pointing this out. The iButtons was placed at the surface, not drilled into the ground, protected by small cairns. We added this to the methods-section. We do not have detailed knowledge of the snow depth in winter or early spring, but expect the top areas to be bare or to have limited snow cover based on the good connection to the SeNorge air temperature date. The lower iButton measurement are more likely to be covered in decimeters of snow by the end of winter. This is also why we feel this should not be interpreted into permafrost/no permafrost without more knowledge of the actual snow conditions in the area.

Table 1

I guess this is in meters.

Yes, you are right. We have corrected the table.

Table 3

I would prefer graph instead of this tabel, it is more expressive and much easier to follow!

It will also be necessary to have a mean multiannual MAAT.

Thank you for this comment. We looked at the table again, and made a corresponding graph, but ended up removing the whole table/graph. We realized that it does not say anything else than what we show in Fig. 7, with the exception of also adding the (modelled) temperatures of the plateau mountain behind the Ivarsfjord rock glacier. The latter temperatures are based on the same regression equation that is shown in Fig. 7 so the two graphs/numbers in the former Table 3 follow each other accordingly. The main point is that the temperature in the area raised over the past 150 years, and that the absolute values are lower at higher altitudes, and we believe this point is communicated in the paper anyway.

On behalf of myself and the co-authors, Karianne S Lilleøren

---

## Editor Decision (ED1)

**Minor technical corrections: Transitional rock glaciers at sea-level in Northern Norway**

Frances Butcher, Associate Editor.

**Technical corrections/suggestions**

Line 158: Please refer to the comment on Table 1 by Reviewer 1 - I do not think this has been corrected. The values in line 158 are inconsistent with those in Table 1 - this appears to be a unit error (column title in Table 1 states cm, but values appear to be in m). Please check the text and Table 1 and correct for this and all techniques that appear in Table 1

Line 152: If 'Unmanned Air Vehicle' is not inherent in the name of the UAV product, I encourage the authors to consider using a gender-neutral equivalent term 'Uncrewed Aerial Vehicle'. Note also Air vs Aerial – I believe aerial is the commonly used word.

Line 255: Is it possible to include absolute displacements in the table to demonstrate the difference between the measurements and the accuracy theshold?

Line 446: It is not clear what 'current framework conditions' means. Do you mean 'in the current framework, conditions are not comparable'?

Table 1: See comment on Line 158: should the unit label for column 2 be m (not cm as it is currently).

Table 2: Please add 'rates' to the Vertical/Horizontal displacement column titles.

Figures:
- Please refer to the ESurf guidelines on figures and tables: https://www.earth-surface-dynamics.net/submission.html#figurestables. In particular, *'Labels of panels must be included with brackets around letters being lower case (e.g. (a), (b), etc.).'*

Figure 1:
- Please check the 'C+D' label in panel C: should these be D+E?

Figure 2:
- In the first map, the map credit is overlain by image 2. Please refer to the ESurf guidelines on the reproduction and reuse of maps and aerials, and ensure that all figures are compliant. https://www.earth-surface-dynamics.net/submission.html#figurestables
- The blue square would benefit from a 'Fig 3' label within the image.

Figure 4:
- The text in these panels is small - I suggest enlarging it, and particularly the legend text. If necessary, the legend could be included below the respective images to allow the text to be larger.
- Basemap images require credit line.

Figure 6:
- Please run the figures through the Coblis colour blindness checker to ensure that figures are accessible, in line with ESurf guidelines: https://www.earth-surface-dynamics.net/submission.html#figurestables https://www.color-blindness.com/coblis-color-blindness-simulator/
- The X referred to in the caption is missing from the image.

Figure 7:
- Could the unit of temperature anomaly be moved to the same line as the text?

**Typographical/grammatical corrections:**

Line 22-23: are no longer permafrost and ground ice > is no longer permafrost or ground ice

Line 32: at least two > for at least two

Line 57: we have had > we had

Line 68: investigate is missing first 'i'.

Line 99: in outer > in the outer

Line 134: Is 'Norway' needed here?

Line 141: Is capitalisation required for 'Stacking'?

Line 161: Delete second occurrence of '(correlation image analysis software)'

Line 180: Put 'ice, water, air and rock' in parentheses.

Line 238: Delete nested parentheses in Malmstrom and Palmer citation.

Line 265: Raise > raising.

Line 366: TLS' > TLSs (apostrophe not appropriate here).

Line 326: in Southern > in the southern.

Line 327-326: Missing close bracket.

Line 335: Interpreted compacted > interpreted as compacted

Line 340: This would be better as: 'The GST monitoring clearly showed annual average temperatures above...'

Line 342: variation > variations

Line 359: than > to

Line 359: facts > fact

Line 360: other processes than > processes other than

Line 365: one information among > one piece of information among

Line 370: they almost exclusively are found > they are almost exclusively found.

Line 394: such e.g. suggested > such as is suggested, for example, for...

Line 425: to footslope landforms > for footslope landforms

Line 426-427: *'Liestøl (1961) acknowledges that the "talus terraces" at numerous locations resemble rock glaciers, and would indeed fall into the present-day rock glacier definition.'* Needs 'they' before 'would'. Otherwise it reads as if Liestol themself acknowledges the features do indeed fall into the present-day definition.

Line 794: are the relict shorelines > the relict shorelines are

---

## Author Response (AR2)

Dear Editor,

Thank you again for considering our paper for publications, and for your careful reading and suggestions that has improved our manuscript.

Below we have addressed all your concerns, and at the end of this document the questions raised by Referee 1.

I hope you will find this satisfactory.

Sincerely, Karianne S Lilleøren (on behalf of all authors).
* * *
Minor technical corrections: Transitional rock glaciers at sea-level in Northern Norway

Frances Butcher, Associate Editor.

**Technical corrections/suggestions**

Line 158: Please refer to the comment on Table 1 by Reviewer 1 - I do not think this has been corrected. The values in line 158 are inconsistent with those in Table 1 - this appears to be a unit error (column title in Table 1 states cm, but values appear to be in m). Please check the text and Table 1 and correct for this and all techniques that appear in Table 1.

This has been corrected. Sorry that this issue remained from the last revision.

Line 152: If 'Unmanned Air Vehicle' is not inherent in the name of the UAV product, I encourage theauthors to consider using a gender-neutral equivalent term 'Uncrewed Aerial Vehicle'. Note also Air vs Aerial – I believe aerial is the commonly used word.

Thank you for noticing, this has been corrected.

Line 255: Is it possible to include absolute displacements in the table to demonstrate the difference between the measurements and the accuracy theshold?

We have added absolute values to the table below, but kept the document table as is was for now. Maybe this is what you requested? If so, it can be replaced with Table 2 in the document text.

**Table 2: Vertical and horizontal displacements in yearly rates (cm/year) and absolute values (cm) of different pairs of DEMs and orthophotos from Ivarsfjorden rock glacier. TLS: Terrestrial Laser Scans.**

| | Vertical displacement | | Horizontal displacement | |
|---|---|---|---|---|
| Period | Range (Mean) [cm yr$^{-1}$] | Range (Mean) [cm] | Range (Mean) [cm yr$^{-1}$] | Range (Mean) [cm] |

| | | | | | |
|---|---|---|---|---|---|
| **1975-1982** | -30 – 10 (-10) | -210 – 70 (-70) | 0 – 2 (1) | 0 – 14 (7) | Historical aerial photos |
| **1975-1992** | -10 – 0 (-5) | -170 – 0 (-50) | 0 – 5 (1) | 0 – 85 (17) | |
| **1975-2017** | -10 – 0 (-7) | -420 – 0 (-294) | 0 – 3 (1) | 0 – 126 (42) | |
| **1982-1992** | -10 – 20 (-5) | -100 – 200 (-50) | 0 – 6 (3) | 0 – 60 (30) | |
| **1982-2017** | -7 – 5 (-2) | -245 – 175 (-70) | 0 – 2 (1) | 0 – 70 (35) | |
| **1992-2017** | -10 – 10 (-3) | -250 – 250 (-75) | 0 – 2 (1) | 0 – 50 (25) | |
| **2016-2017** | -10 – 5 (0) | -10 – 5 (0) | 0 – 3 (2) | 0 – 3 (2) | Drone photos |
| **2017-2020** | -5 – 2 (-1) | -15 – 6 (-3) | 0 – 3 (2) | 0 – 9 (6) | TLS |

Line 446: It is not clear what 'current framework conditions' means. Do you mean 'in the current framework, conditions are not comparable'?

We have changed the text to your suggestion.

Table 1: See comment on Line 158: should the unit label for column 2 be m (not cm as it is currently).

Yes, this has been changed now.

Table 2: Please add 'rates' to the Vertical/Horizontal displacement column titles.

We have added "rates" to the column titles.

**Figures:**

• Please refer to the ESurf guidelines on figures and tables: https://www.earth-surfacedynamics.net/submission.html#figurestables. In particular, 'Labels of panels must be included with brackets around letters being lower case (e.g. (a), (b), etc.).'

This has been corrected.

Figure 1:

• Please check the 'C+D' label in panel C: should these be D+E?

Sorry, our mistake. This has been corrected.

Figure 2:

• In the first map, the map credit is overlain by image 2. Please refer to the ESurf guidelines on the reproduction and reuse of maps and aerials, and ensure that all figures are compliant. https://www.earth-surface-dynamics.net/submission.html#figurestables

We have moved the image accordingly.

• The blue square would benefit from a 'Fig 3' label within the image.

Fig. 3 has been added to the image.

Figure 4:

• The text in these panels is small - I suggest enlarging it, and particularly the legend text. If necessary, the legend could be included below the respective images to allow the text to be larger.

We have increased the panels and text.

• Basemap images require credit line.

We have also added the credit line for the background orthophotos.

Figure 6:

• Please run the figures through the Coblis colour blindness checker to ensure that figures are accessible, in line with ESurf guidelines: https://www.earth-surfacedynamics.net/submission.html#figurestables

https://www.color-blindness.com/coblis-color-blindness-simulator/

We have run the figure through this color blindness tool, thank you so much for pointing at this. To my understanding, this figure is accessible for all the simulated color blindness tests.

• The X referred to in the caption is missing from the image.

The X is now back in the figure.

Figure 7:

• Could the unit of temperature anomaly be moved to the same line as the text?

Yes, it has been moved to the text line.

**Typographical/grammatical corrections:**

**All of the following corrections have been changed.**

Line 22-23: are no longer permafrost and ground ice > is no longer permafrost or ground ice

Line 32: at least two > for at least two

Line 57: we have had > we had

Line 68: investigate is missing first 'i'.

Line 99: in outer > in the outer

Line 134: Is 'Norway' needed here?

Line 141: Is capitalisation required for 'Stacking'?

Line 161: Delete second occurrence of '(correlation image analysis software)'

Line 180: Put 'ice, water, air and rock' in parentheses.

Line 238: Delete nested parentheses in Malmstrom and Palmer citation.

Line 265: Raise > raising.

Line 366: TLS' > TLSs (apostrophe not appropriate here).

Line 326: in Southern > in the southern.

Line 327-326: Missing close bracket.

Line 335: Interpreted compacted > interpreted as compacted

Line 340: This would be better as: 'The GST monitoring clearly showed annual average temperatures above…'

Line 342: variation > variations

Line 359: than > to

Line 359: facts > fact

Line 360: other processes than > processes other than

Line 365: one information among > one piece of information among

Line 370: they almost exclusively are found > they are almost exclusively found.

Line 394: such e.g. suggested > such as is suggested, for example, for...

Line 425: to footslope landforms > for footslope landforms

Line 426-427: 'Liestøl (1961) acknowledges that the "talus terraces" at numerous locations resemble rock glaciers, and would indeed fall into the present-day rock glacier definition.' Needs 'they' before 'would'. Otherwise it reads as if Liestol themself acknowledges the features do indeed fall into the present-day definition.

Line 794: are the relict shorelines > the relict shorelines
* * *
Comments from Referee 1.

Dear Referee 1,

Thank you for your comments and suggestions. We have addressed your points below, and I am sorry for those that remained from the last version of the manuscript.

I hope this is satisfactory.

Sincerely, Karianne S Lilleøren (on behalf of all authors)

1) In the paragraph where you give some details on the thermal camera (line 225 in the track-change version of the manuscript), it is still not clear what "systematically" means. Did you measure the front only once or repeatedly? Every year? What about the weather conditions and their influences?

L194 and L297: We took pictures one day in September, 2018. This day was unusually hot, and we felt gusts of cold air escaping the front of the rock glacier. I should probably exclude the word "systematically", since this was only performed one year.

The text has been changed to (L195ff): "(…) we investigated the rock glacier front using a thermal camera (Teledyne FLIR C3) measuring infrared radiation. On one day in September 2018, we took ca. 20 pictures distributed along the front, with a 1 m distance between the camera and the object."

2) Figure 1: in the permafrost map it still looks like two different pinkish colors (one for possible and one for probable permafrost?).

This was two colors in the previous version of the manuscript, but in this version there is only one purple color. It may be the background map that disturbs the visibility a little.

3) Figure 8: following the figure caption, the labeling was not adapted in the figure (A for the temperature plot, ... D for the overview image).

Thank you for noticing, this has now been corrected.